# CytoCensus, mapping cell identity and division in tissues and organs using machine learning

Martin Hailstone[1], Dominic Waithe[2], Tamsin J Samuels[1], Lu Yang[1], Ita Costello[3], Yoav Arava[4], Elizabeth Robertson[3], Richard M Parton[1,5], Ilan Davis[1,5]*

[1]Department of Biochemistry, University of Oxford, Oxford, United Kingdom; [2]Wolfson Imaging Center & MRC WIMM Centre for Computational Biology MRC Weather all Institute of Molecular Medicine University of Oxford, Oxford, United Kingdom; [3]The Dunn School of Pathology,University of Oxford, Oxford, United Kingdom; [4]Department of Biology, Technion - Israel Institute of Technology, Haifa, Israel; [5]Micron Advanced Bioimaging Unit, Department of Biochemistry, University of Oxford, Oxford, United Kingdom

**Abstract** A major challenge in cell and developmental biology is the automated identification and quantitation of cells in complex multilayered tissues. We developed CytoCensus: an easily deployed implementation of supervised machine learning that extends convenient 2D 'point-and-click' user training to 3D detection of cells in challenging datasets with ill-defined cell boundaries. In tests on such datasets, CytoCensus outperforms other freely available image analysis software in accuracy and speed of cell detection. We used CytoCensus to count stem cells and their progeny, and to quantify individual cell divisions from time-lapse movies of explanted *Drosophila* larval brains, comparing wild-type and mutant phenotypes. We further illustrate the general utility and future potential of CytoCensus by analysing the 3D organisation of multiple cell classes in Zebrafish retinal organoids and cell distributions in mouse embryos. CytoCensus opens the possibility of straightforward and robust automated analysis of developmental phenotypes in complex tissues.

*For correspondence:
ilan.davis@bioch.ox.ac.uk

## Introduction

Complex tissues develop through regulated proliferation and differentiation of a small number of stem cells. For example, in the brain these processes of proliferation and differentiation lead to a vast and diverse population of neurons and glia from a limited number of neural stem cells, also known as neuroblasts (NBs) in *Drosophila* (*Kohwi and Doe, 2013*). Elucidating the molecular basis of such developmental processes is not only essential for understanding basic neuroscience but is also important for discovering new treatments for neurological diseases and cancer. Modern imaging approaches have proven indispensable in studying development in intact zebrafish (*Danio rario*) and *Drosophila* tissues (*Barbosa and Ninkovic, 2016*; *Dray et al., 2015*; *Medioni et al., 2015*; *Rabinovich et al., 2015*; *Cabernard and Doe, 2013*; *Graeden and Sive, 2009*). Tissue imaging approaches have also been combined with functional genetic screens, for example to discover NB behaviour underlying defects in brain size or tumour formation (*Berger et al., 2012*; *Homem and Knoblich, 2012*; *Neumüller et al., 2011*). Such screens have the power of genome-wide coverage, but to be effective, require detailed characterisation of phenotypes using image analysis. Often these kinds of screens are limited in their power by the fact that phenotypic analysis of complex tissues can only be carried out using manual image analysis methods or complex bespoke image analysis.

**eLife digest** There are around 200 billion cells in the human brain that are generated by a small pool of rapidly dividing stem cells. For the brain to develop correctly, these stem cells must produce an appropriate number of each type of cell in the right place, at the right time. However, it remains unclear how individual stem cells in the brain know when and where to divide.

To answer this question, Hailstone et al. studied the larvae of fruit flies, which use similar genes and mechanisms as humans to control brain development. This involved devising a new method for extracting the brains of developing fruit flies and keeping the intact tissue alive for up to 24 hours while continuously imaging individual cells in three dimensions.

Manually tracking the division of each cell across multiple frames of a time-lapse is extremely time consuming. To tackle this problem, Hailstone et al. created a tool called CytoCensus, which uses machine learning to automatically identify stem cells from three-dimensional images and track their rate of division over time. Using the CytoCensus tool, Hailstone et al. identified a gene that controls the diverse rates at whichstem cells divide in the brain. Earlier this year some of the same researchers also published a study showing that this gene regulates a well-known cancer-related protein using an unconventional mechanism.

CytoCensus was also able to detect cells in other developing tissues, including the embryos of mice. In the future, this tool could aid research into diseases that affect complex tissues, such as neurodegenerative disorders and cancer.

*Drosophila* larval brains develop for more than 120 h (*Homem and Knoblich, 2012*), a process best characterised by long-term time-lapse microscopy. However, to date, imaging intact developing live brains has tended to be carried out for relatively short periods of a few hours (*Lerit et al., 2014*; *Cabernard and Doe, 2013*; *Prithviraj et al., 2012*) or using disaggregated brain cells in culture (*Homem et al., 2013*; *Moraru et al., 2012*; *Savoian and Rieder, 2002*; *Furst and Mahowald, 1985*). Furthermore, although extensively studied, a range of different division rates for both NBs and progeny ganglion mother cells (GMCs) are reported in the literature (*Homem et al., 2013*; *Bowman et al., 2008*; *Ceron et al., 2006*) and in general, division rates have not been systematically determined for individual neuroblasts. Imaging approaches have improved rapidly in speed and sensitivity, making imaging of live intact tissues in 3D possible over developmentally relevant timescales. However, long-term exposure to light often perturbs the behaviour of cells in subtle ways. Moreover, automated methods for the analysis of the resultant huge datasets are still lagging behind the microscopy methods. These imaging and analysis problems limit our ability to study NB development in larval brains, as well as more generally our ability to study complex tissues and organs.

Here, we describe our development and validation of *ex vivo* live imaging of *Drosophila* brains, and of CytoCensus, a machine learning-based automated image analysis software that fills the technology gap that exists for images of complex tissues and organs where segmentation and spot detection approaches can struggle. Our program efficiently and accurately identifies cell types and divisions of interest in very large (50 GB) multichannel 3D and 4D datasets, outperforming other state-of-the-art tools that we tested. We demonstrate the effectiveness and flexibility of CytoCensus first by quantitating cell type and division rates in *ex vivo* cultured intact developing *Drosophila* larval brains imaged at 10% of the normal illumination intensity with image quality restoration using patched-based denoising algorithms (*Carlton et al., 2010*). Second, we quantitatively characterise the precise numbers and distributions of the different cell classes within two vertebrate tissues: 3D Zebrafish organoids and mouse embryos. In all these cases, CytoCensus successfully outputs quantitation of the distributions of most cells in tissues that are too large or complex for practical manual annotation. Our software provides a convenient tool that works 'out-of-the-box' for quantitation and single-cell analysis of complex tissues in 4D, and, in combination with other software (e.g. FIJI), supports the study of more complex problems than would otherwise be possible. CytoCensus offers a practical alternative to producing bespoke image analysis pipelines for specific applications.

## Motivation and design

We sought to overcome the image analysis bottleneck that exists for complex tissues and organs by creating easy to use, automated image analysis tools able to accurately identify cell types and determine their distributions and division rates in 3D, over time within intact tissues. To date, challenging image analysis tasks of this sort have largely depended on slow, painstaking manual analysis, or the bespoke development or modification of dedicated specialised tools by an image analyst with significant programming skills (*Chittajallu et al., 2015*; *Schmitz et al., 2014*; *Stegmaier et al., 2014*; *Homem et al., 2013*; *Myers, 2012*; *Meijering, 2012*; *Meijering et al., 2012*; *Rittscher, 2010*). Of the current freely available automated tools, amongst the most powerful are Ilastik and the customised pipelines of the FARSIGHT toolbox and CellProfiler (*Padmanabhan et al., 2014*; *Sommer and Gerlich, 2013*; *Sommer, 2011*; *Roysam et al., 2008*). However, these three approaches require advanced knowledge of image processing, programming and/or extensive manual annotation. Other software such as Advanced Cell Classifier are targeted at analysis of 2D data, whilst programs such as RACE, SuRVoS, 3D-RSD and MINS are generally tailored to specific applications (*Luengo et al., 2017*; *Stegmaier et al., 2016*; *Lou et al., 2014*; *Cabernard and Doe, 2013*; *Homem et al., 2013*; *Arganda-Carreras et al., 2017*; *Logan et al., 2016*; *Gertych et al., 2016*). Recently, efforts to make deep learning approaches easily accessible have made great strides (*Falk et al., 2019*); such implementations have the potential to increase access to these powerful supervised segmentation methods, but at present hardware and installation requirements are likely to be too complex for the typical biologist. In general, we find that existing tools can be powerful in specific examples, but lack the flexibility, speed and/or ease of use to make them effective solutions for most biologists in the analysis of large time-lapse movies of 3D developing tissues.

In developing CytoCensus, we sought to design a widely applicable, supervised machine leaning-based image analysis tool, addressing the needs of biologists to efficiently characterise and quantitate dense complex 3D tissues at the single-cell level with practical imaging conditions. This level of analysis of developing tissues, organoids or organs is frequently difficult due to the complexity and density of the tissue arrangement or labelling, as well as limitations of signal to noise. We therefore aimed to make CytoCensus robust to these issues but also to make it as user friendly as possible. In contrast to other image analysis approaches that require the user to define the cell boundaries, CytoCensus simply requires the user to point-and-click on the approximate centres of cells. This single click training need only be carried out on a few representative 2D planes from a large 3D volume, and tolerates relatively poor image quality compatible with extended live cell imaging. To make the task very user friendly, we preconfigured most algorithm settings leaving a few, largely intuitive, parameters for the user to set. To improve performance, we enabled users to define regions of interest (ROIs) which exclude parts of a tissue that are not of interest or interfere with the analysis. We also separated the training phase from the analysis phase, allowing efficient batch processing of data. For the machine learning, we choose a variation of Random Forests with pre-calculated image features, which allows for much faster training compared to neural networks on typical computers, and with a fraction of the user annotation. A similar approach is taken by the image analysis software Ilastik (*Berg et al., 2019*). Using machine learning, CytoCensus then determines the probability of each pixel in the image being the centre of the highlighted cell class in 3D, based on the characteristics of the pixels around the site clicked. This proximity map is used to identify all of the cells of interest. Finally, to increase the ease of adoption, we designed CytoCensus to be easily installed and work on multiple platforms and computers with standard specifications, including generically configured laptops without any pre-requisites. Collectively, these improvements make CytoCensus an accessible and user-friendly image analysis tool that will enable biologists to analyse their image data effectively, increase experimental throughput and increase the statistical strength of their conclusions.

## Results

### Optimised time-lapse imaging of developing intact *ex vivo* brains

To extend our ability to study stem cell behaviour in the context of the intact *Drosophila* brain, we modified the methods of *Cabernard and Doe (2013)*, revised in *Syed et al. (2017)*, to produce a convenient and effective protocol optimising tissue viability for long-term culture and quantitative

imaging. We first developed an isolation procedure incorporating scissor-based dissection of second or third-instar larvae, in preference to solely tweezer or needle-based dissection which can damage the tissue. We then simplified the culture medium and developed a convenient brain mounting technique that immobilises the organ using agar (*Figure 1A*; Materials and methods). We also made use of bright, endogenously expressed fluorescently tagged proteins Jupiter::GFP and Histone::RFP marking microtubules and chromosomes respectively, to follow the developing brain (*Figure 1B*). We chose generic cytological markers as these are more consistent across *wild-type* (WT) and different mutants than more specific markers, such as Deadpan (Dpn), Asense (Ase) or Prospero (Pros), commonly used to identify NBs, GMCs and neurons. Finally, we optimised the imaging conditions to provide 3D data sets of sufficient temporal and spatial resolution to follow cell proliferation over time without compromising viability (see Materials and methods). Significantly, to maximise temporal and spatial resolution without causing damage, we reduced photo-damage by decreasing the laser excitation power by approximately 10 fold (see Materials and methods) and subsequently restoring image quality using patch-based denoising (*Carlton et al., 2010*), developed by *Kervrann and Boulanger (2006)*. This approach allowed us to follow the lineage and quantitate the divisions of NBs and GMCs in the intact brain in 3D (*Figure 1C,D*).

To assess whether our culturing and imaging protocol supports normal development, we used a number of criteria. We found that by all the criteria we measured, brain development is normal in our *ex vivo* conditions. First, the cultured *ex vivo* brains do not show signs of damage during preparation, which can be easily identified as holes or lesions in the tissue that expand with time in culture. Second, our cultured larval brains consistently increase in size as they progress through development (*Figure 1—figure supplement 1*). Third, using our approach, we recorded average division rates of 0.66 divisions/hour (~90 min per cycle, *Figure 1C*) for the Type 1 NB of the central brain (*Figure 1— figure supplement 1A'*), at the wandering third instar larval stage (wL3), as previously published (*Homem et al., 2013*; *Bowman et al., 2008*; *Figure 1—video 1*; *Figure 4—video 1*). We note here that experiments were performed at 21°C, which differs from some developmental studies performed at 25°C. Type I NBs were identified by location according to *Homem and Knoblich (2012)*. Fourth, we rarely observed excessive lengthening or arrest of the cell cycle in NBs over a 22 h imaging period, which is approximately the length of the wL3 stage (*Figure 1C*). With longer duration culture and imaging, up to 48 hr, we observe an increase in cell cycle length, which might be expected for wL3 brains transitioning to the pupal state (*Homem et al., 2014*). Finally, we observed normal and sustained rates of GMC division throughout the imaging period that correspond to the previously described literature in fixed brain preparations (*Bowman et al., 2008*; *Figure 1D*; *Figure 4— video 2*). We conclude that our *ex vivo* culture and imaging methods accurately represent development of the *Drosophila* brain and support high time and spatial resolution imaging for quantitation of cell numbers and division rates.

## CytoCensus enables easy automated quantification of cell types in time-lapse movies of developing intact larval brains with modest training

Progress in elucidating the molecular mechanisms of regulated cell proliferation during larval brain development has largely depended on the characterisation and quantification of mutant phenotypes by painstaking manual image analysis (for example, *Neumüller et al., 2011*). However, the sheer volume of image data produced by whole brain imaging experiments means that manual assessment is impractical. Therefore, we attempted to use freely available image analysis tools in an effort to automate the identification of cell types. We found that none of the available off-the-shelf image analysis programs perform adequately on our complex 3D datasets, in terms of ease of use, speed or accuracy (*Table 1*). Neuroblast nuclei are large and diffuse, which means that conventional spot detectors (e.g. TrackMate) struggle to identify them. Ilastik Density Counting (which takes a related approach to CytoCensus) was promising to count NB in 2D, but is not designed to work in 3D nor to detect cell centres (*Berg et al., 2019*). Similarly, image segmentation tools (such as RACE, Ilastik Pixel Classification and WEKA) struggle to segment NB marked by microtubule labels as they vary significantly in appearance with the cell cycle and cell boundaries may appear incomplete. To overcome these limitations, we developed CytoCensus, an easily deployed, supervised machine learning-based image analysis software (*Figure 2*; *Figure 2—figure supplement 1*). CytoCensus facilitates automated detection of cell types and quantitative analysis of cell number, distribution and proliferation

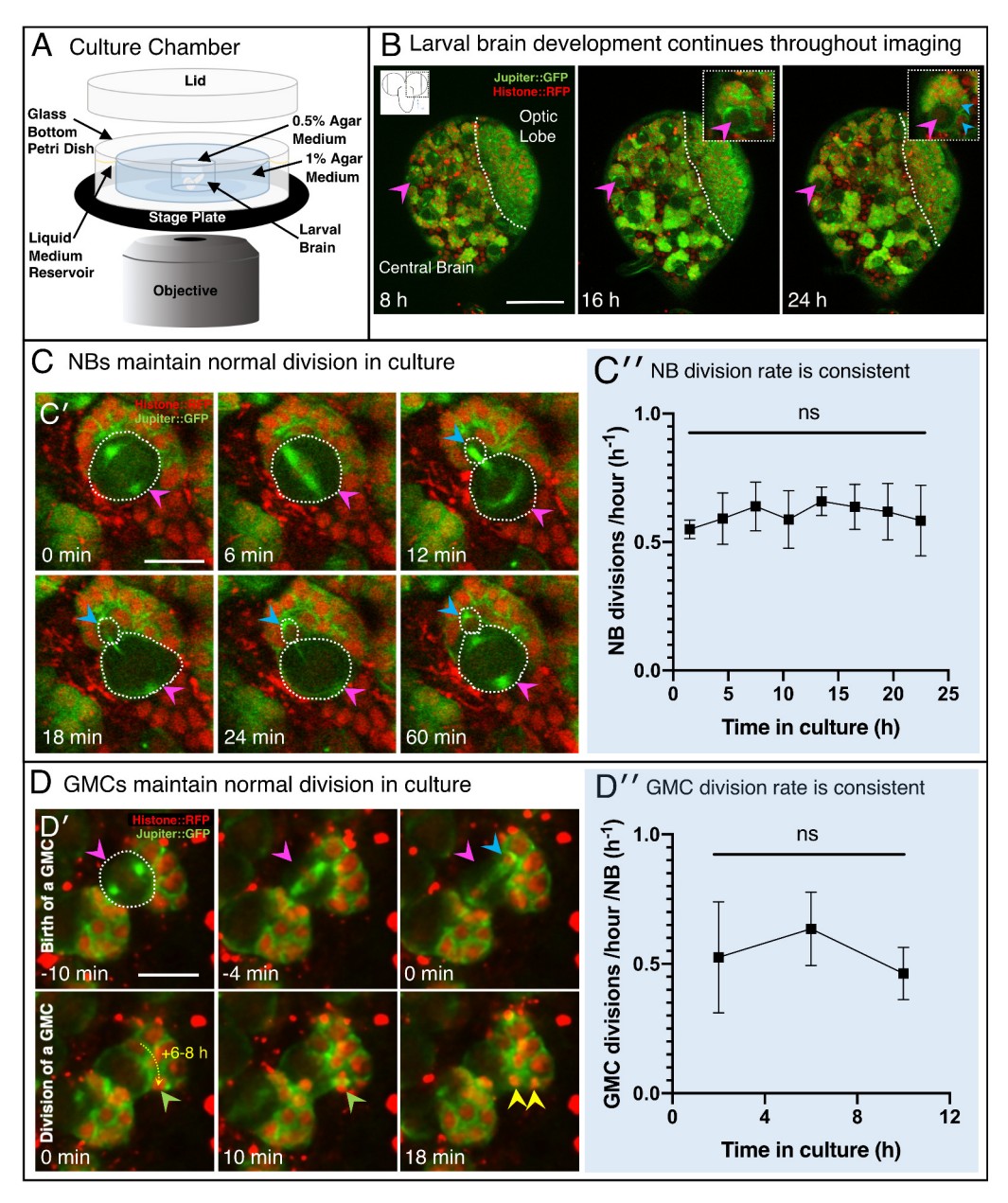

**Figure 1.** Extended 3D time-lapse imaging of live *ex vivo* cultured brains. (**A**) Diagram of the chamber and sample preparation for long-term time-lapse imaging on an inverted microscope (see Materials and methods). (**B**) 24 h, confocal 3D time-lapse imaging of a developing larval brain lobe (inset, top left, shows orientation and region of the brain imaged) labelled with Jupiter::GFP and Histone::RFP, and registered over time to account for movement. Arrowheads indicate NBs (magenta) and progeny (cyan), enlarged in the top right insets; a dashed white line indicates the boundary to the optic lobe. (**C′**) A typical individual dividing NB from a confocal time-lapse image sequence of the brain lobe. The NB is outlined (dashed white line) and indicated with a magenta arrowhead, the progeny (GMC) is indicated by a cyan arrowhead. (**C″**) Plot of NB division rate for cultured L3 brains shows that division rate of NBs does not significantly decrease over at least 22 h under imaging conditions (n = 3 brains, not significant (ns), p=0.87, one-way ANOVA), calculated from measured cell cycle lengths. (**D′**) Typical GMC division in an intact larval brain. The first row of panels shows production of a GMC (cyan arrowhead) by the dividing NB (magenta arrowhead, dashed white outline). Second row of panels, GMCs are displaced over the next 6 to 8 h by subsequent NB divisions, the path of displacement is indicated by the dashed yellow arrow. The last two panels (10 to 18 min) show the division of a GMC (green arrowhead, progeny yellow arrowheads). (**D″**) Plot showing the rate of GMC division in the *ex vivo* brain does not change with time in culture (n = 4 brains, ns, p=0.34, one-way ANOVA), calculated from the number of GMC division events in 4 h. Error bars on plots are standard deviation. Scale bars (**B**) 50 μm; (**C**), (**D**) 10 μm.

The online version of this article includes the following video and figure supplement(s) for figure 1:

**Figure supplement 1.** *Ex vivo* larval brains continue to develop in culture.

*Figure 1 continued on next page*

*Figure 1 continued*
**Figure 1—video 1.** Development of a live explanted larval brain under extended time-lapse imaging conditions.
https://elifesciences.org/articles/51085#fig1video1

from time-lapse movies of multichannel 3D image stacks even in complex tissues. A full technical description of the algorithm is found in Materials and methods and a User Guide is available in the Supplemental Information.

To optimise its effectiveness, we developed CytoCensus with a minimal requirement for supervision during the training process. We developed an implementation of supervised machine learning (see Materials and methods), in which the user trains the program in 2D on a limited number of images (*Figure 2*). In this approach, the user simply selects, with a single mouse click, the approximate centres of all examples of a particular cell type within small user-defined regions of interest in the image. This makes CytoCensus is more convenient and faster than other machine learning-based approaches, such as FIJI-WEKA (*Arganda-Carreras et al., 2017*) or Ilastik Pixel Classification, (*Sommer, 2011*), which require relatively extensive and time consuming annotation of the cells by their boundaries. However, this simple training regime requires assumptions of roundness, which precludes direct analysis of cell shape. We explore the extent of this limitation in subsequent sections.

To further optimise the training, our training workflow outputs a 'proximity' map, similar to those described in *Fiaschi et al. (2012)*; *Swiderska-Chadaj et al. (2018)*; *Liang et al. (2019)*; *Höfener et al. (2018)*. These approaches all focus on 2D proximity maps, while CytoCensus utilises proximity maps in 3D. One may think of the proximity map as a probability of how likely it is that a given pixel is at the center of one of the cells of interest. Using this proximity map the user can assess the accuracy of the prediction and, if necessary, provide additional training (*Figure 2*). This proximity map and the predicted locations of cell centres across the entire volume and time-series are saved and may be conveniently passed to ImageJ (FIJI), or other programs (*Schindelin et al., 2012*) for further processing (*Figure 2*; *Figure 2—figure supplement 1*; *Figure 3—figure supplement 1*). After this initial phase of manual user training, the subsequent processing of new unseen data is automated and highly scalable to large image data sets without any further manual user training. To determine the required training, the impact of training level (number of regions used in the training) was assessed on live imaging data sets (Materials and methods). The results show that detection accuracy was optimised even with a modest levels of training (*Figure 3—figure supplement 2*).

## CytoCensus is a significant advance in automated cell detection in challenging data sets

We assessed the performance of CytoCensus at cell identification on challenging live imaging data sets that were manually annotated by a user to generate 'ground-truth' results. Before comparison between applications, algorithm parameters were optimised for the different approaches to prevent overfitting (Materials and methods). In our tests CytoCensus outperformed the machine learning based approaches Fiji-WEKA (p=0.005, t-test, n = 3) and Ilastik Pixel Classification (p=0.007, t-test, n = 3), and other freely available approaches in the accuracy of NB detection, speed and simplicity of use (*Figure 3A*; *Table 1*). We calculated a metric of performance, intuitively similar to accuracy, which is known as the F1-score, with a maximum value of 1.0 (Materials and methods; *Table 1*). We found that the best performing approaches on our complex datasets were Ilastik Pixel Classification and CytoCensus, which are machine learning based. It is likely that both approaches might be further improved with additional bespoke analysis, specific to each data set, however this would limit their flexibility and ease of use.

To further critically assess the performance of CytoCensus, we used an artificially generated 'neutral challenge' 3D dataset, which facilitates fair comparison (*Figure 3B*). We used a dataset of 30 images of highly clustered synthetic cells, in 3D, with a low signal-to-noise ratio (SNR), obtained from the Broad Bioimage Benchmark Collection (Materials and methods). We selected this dataset because it has similar characteristics to our live imaging data. Using this dataset, we directly compared the abilities of Ilastik Pixel Classification (*Figure 3B''*) and CytoCensus (*Figure 3B'''*), to

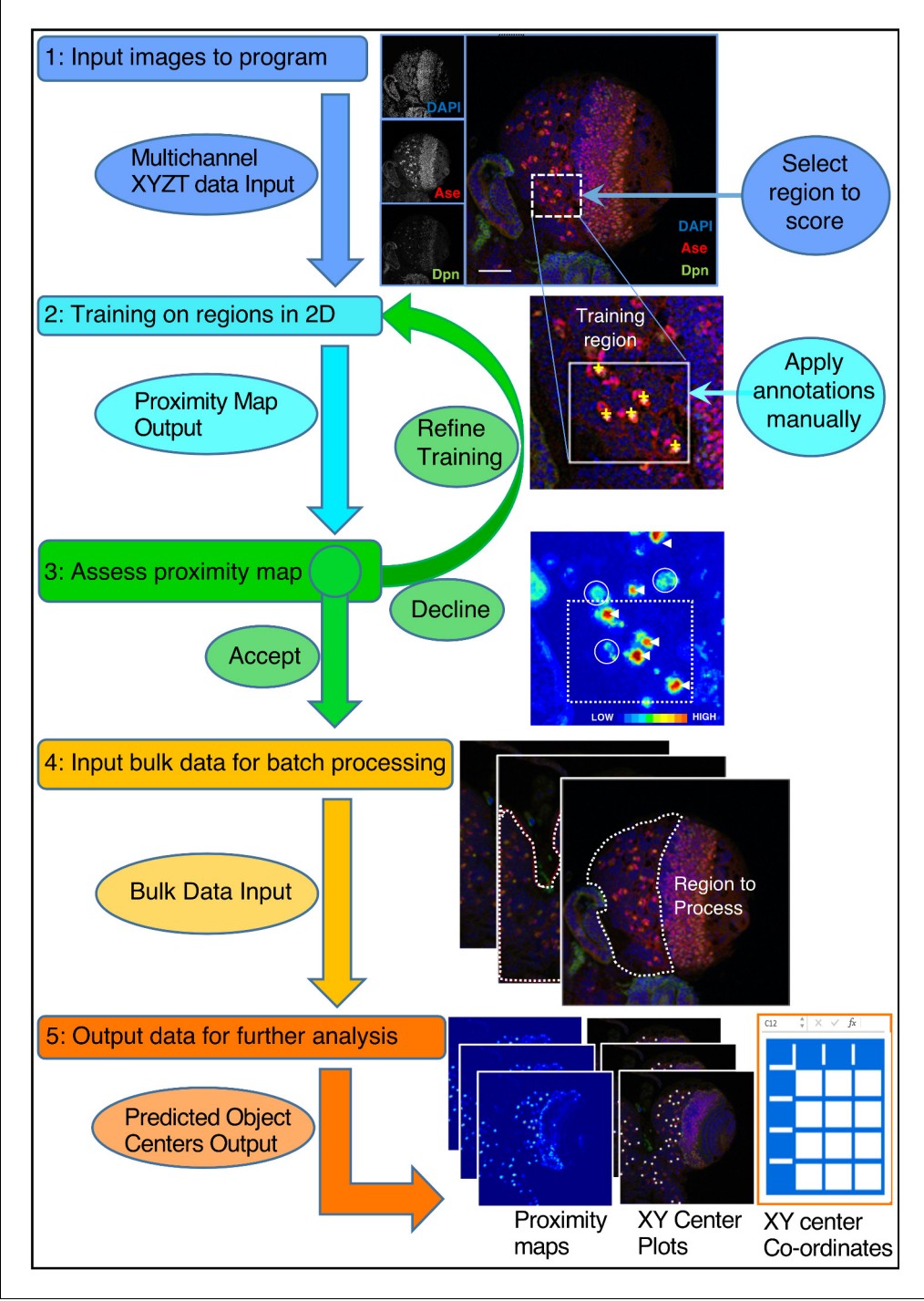

**Figure 2.** CytoCensus analysis workflow. Refer to the Main Text and Materials and methods for details. Training is performed by single click annotation (yellow crosses) within a user-defined region of interest (ROI, white dashed square) to identify the cell class of interest. The resultant proximity map for cell class identification (~probability score for object centres) is evaluated manually to assess the success of training (white arrows indicate good detections and circles indicate where more training may be required). A successful identification regime (Model) is saved and may be used to batch process multiple image data sets. Multiple outputs are produced including a list of the co-ordinates of identified cells. Multiple identification regimes can be sequentially applied to identify multiple cell classes from a single data set.

The online version of this article includes the following figure supplement(s) for figure 2:

**Figure supplement 1.** CytoCensus graphical user interface.

**Table 1.** CytoCensus outperforms other freely available programs for cell class identification.
Performance assessment for a series of freely available tools in identifying NBs from a typical 4D live-imaging time-series of the generic cytological markers Jupiter::GFP/Histone::RFP, expressed in larval brains. Comparison to CytoCensus is made on the same computer, including time taken to provide user annotations for a standard data set (150 or 35 time-points, 30-Z). Computer specifications: MacBook Pro11,5; Intel Core i7 2.88 GHz; 16 GB RAM. For manual annotations, the time taken to annotate the full dataset was estimated from the time to annotate 10 time-points. Values ± standard deviations are shown, n = 3. Fiji, ImageJ V1.51d (*Schindelin et al., 2012*); FIJI, local threshold V1.16.4 (http://imagej.net/Auto_Local_Threshold); FIJI-WEKA, WEKA 3.2.1 (*Arganda-Carreras et al., 2017*); RACE (*Stegmaier et al., 2016*); TrackMate (*Tinevez et al., 2017*); Ilastik (V1.17) (*Logan et al., 2016*; *Sommer, 2011*).

| | Manual | Fiji/auto local threshold | TrackMate spot detection | RACE | Ilastik Pixel Classif-ication (1.17) | Fiji WEKA | Cyto-Census V0.1 |
|---|---|---|---|---|---|---|---|
| Total Parameters to select | - | **1** | **4** | 8 | 67 (48) | 25 | 6 |
| Handles 4-D easily | - | NO | **YES** | **YES** | **YES** | NO | **YES** |
| Time to Train model (min.) | - | N/A | N/A | N/A | 15 | 18 | **6** |
| Time to Run (min. including postprocessing) | 550 (equivalent) | **5** | **1** | 16 | 70 | 105 | 19 |
| F1-score | - | Fail | 0.11 ±0.09 | 0.17 ±0.01 | **0.76 ±0.01** | 0.62 ±0.07 | **0.96 ±0.01** |

identify cell centres in 3D. In both cases, we trained on a single image, optimised parameters on five images, and evaluated performance on the remaining 25 images. We found that CytoCensus (*Table 2*, F1-score: 0.98 ± 0.05) outperforms Ilastik Pixel Classification in the accuracy of cell centre detection (*Figure 3B*) even after the Ilastik Pixel Classification results were post-processed to aid separation of touching objects (*Table 2*, F1-score: 0.88 ± 0.09). We conclude that CytoCensus is significantly more accurate than Ilastik at identifying cells when both are tested out-of-the-box on neutral challenge data (*Figure 3B''''* p<0.001, Welch's t-test, n = 25).

For a more general comparison to other detection and segmentation methods, we applied CytoCensus to 3D data from the Cell Tracking Challenge Segmentation Benchmark (*Ulman et al., 2017*; *Maška et al., 2014*). To properly participate in this challenge, we trained CytoCensus as normal, and then applied a simple post-processing of the CytoCensus centres, using a small number of iterations of MorphACME (*Marquez-Neila et al., 2013*), an active contour segmentation method, in order to get a segmentation. The results of CytoCensus are shown in *Figure 3—figure supplement 2B*,

**Table 2.** CytoCensus outperforms other freely available programs for cell class identification.
Direct comparison of Ilastik Pixel Classification vs CytoCensus in automatically identifying cell centres in a crowded 3D data set. To facilitate fair comparison, a 'neutral challenge dataset' was used (Main Text). F1 score is intuitively similar to accuracy of detection. Values ± standard deviations are shown, n = 25 images. Computer specifications: MacBook Pro11,5; Intel Core i7 2.88 GHz; 16 GB RAM. Ilastik (V1.17) (*Logan et al., 2016*; *Sommer, 2011*).

| | Ilastik pixel classification (1.17) (raw) | Ilastik pixel classification (1.17) (post-processed) | CytoCensus V0.1 |
|---|---|---|---|
| CPU time (hours) | 82 | 83 | **12** |
| Precision (True Positive Rate) | 0.39 ± 0.19 | 0.86 ± 0.10 | **0.98 ± 0.05** |
| Recall (Positive Predictive Value) | 0.15 ± 0.10 | 0.90 ± 0.07 | **0.98 ± 0.05** |
| F1-score (max = 1.0) | 0.21 ± 0.13 | 0.88 ± 0.09 | **0.98 ± 0.05** |

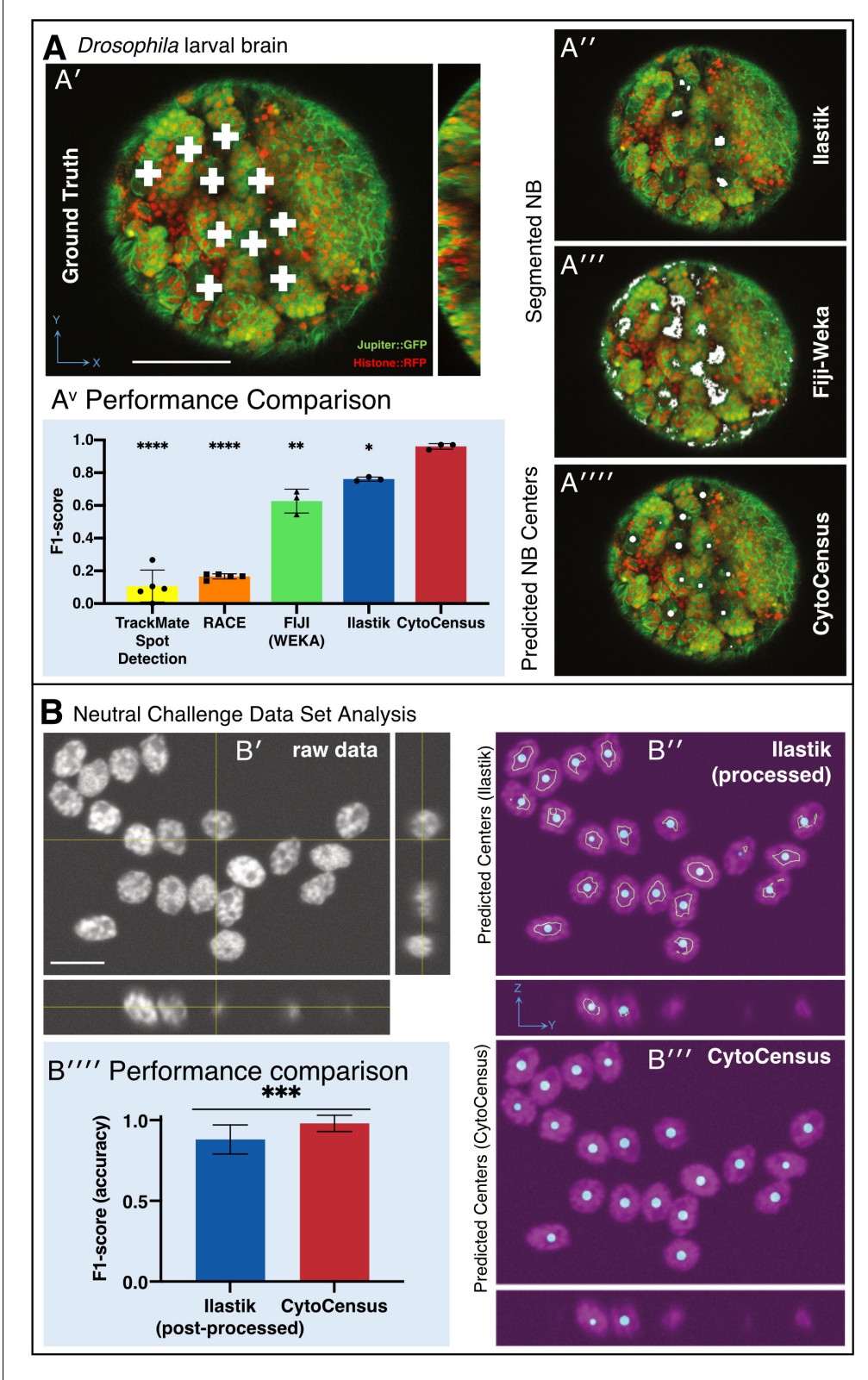

**Figure 3.** Validation of CytoCensus performance. (**A**) Performance in identifying NBs from 3D confocal image data of a live brain labelled with Jupiter::GFP, Histone::RFP. (**A′**) Ground Truth manual identification of NB centres. A′′ to ′′′′) Output images comparing NB identification by Ilastik, Fiji-Weka and CytoCensus, white overlay. (Av) Plot comparing object centre detection by TrackMate spot detection, RACE, Fiji-Weka, Ilastik Pixel Classification and

*Figure 3 continued on next page*

*Figure 3 continued*

CytoCensus (error bars are standard deviation). CytoCensus achieves a significantly better F1-score than Ilastik (p=0.01, n = 3) and FIJI (p=0.005, n = 3). (one-way RM-ANOVA with post hoc t-tests) (B) Comparison of algorithm performance for a 3D neutral challenge data set (B′, see Materials and methods). (B″, B‴) Output images comparing object centre determination by Ilastik Pixel Classification and CytoCensus. Segmentation results are shown as green outlines, object centre determination is show as a cyan point. (B‴′) Plot comparing object centre determination accuracy for the 3D neutral challenge dataset (error bars are standard deviation; p<0.001, Welch's t-test, n = 25). Scale bars (B) 20 μm; (A′) 50 μm.

The online version of this article includes the following figure supplement(s) for figure 3:

**Figure supplement 1.** CytoCensus identification of cell types.

**Figure supplement 2.** Comparison of CytoCensus performance.

along with the results of the top 3 ranked algorithms at the time of publication. CytoCensus in general performs well at detection, achieving top 3 performance on 3/6 datasets tested (*Figure 3—figure supplement 2B″*). CytoCensus with post-processing performs surprisingly well at the segmentation aspect of the challenge, despite primarily being aimed at detection, achieving top 3 in 2/6 datasets (*Figure 3—figure supplement 2B‴*). Unsurprisingly, datasets such as N3DH-CE, which have cells with highly variable cell size and shape, were challenging for CytoCensus. However, CytoCensus performed particularly well on datasets with lower signal to noise and a high density of cells (e.g. N3DH-SIM+, N3DH-DRO, N3DH-TRIC [*Jain et al., 2019*]), illustrating where CytoCensus is best applied. In general CytoCensus performs competitively on the tested datasets, leveraging a small amount of training to achieve good results without needing dataset specific algorithms for denoising or object separation.

We conclude that CytoCensus represents a significant advance over the other current freely available methods of analysis, both in ease of use and in ability to accurately and automatically analyse cells of interest in the large volumes of data resulting from live imaging of an intact complex tissue such as a brain. This will greatly facilitate the future study of subtle or complex mutant developmental phenotypes.

## Using CytoCensus to analyse the over-growth phenotype of *syncrip* knockdown larval brains

To demonstrate the power and versatility of using CytoCensus in the analysis of a complex brain mutant phenotype, we characterised the brain overgrowth phenotype of *syncrip* (*syp*) knockdown larvae (*Figure 4A*). SYNCRIP/hnRNPQ, the mammalian homologue of Syp, is a component of RNA granules in the dendrites of mammalian hippocampal neurons (*Bannai et al., 2004*). Syp also determines neuronal fate in the *Drosophila* brain (*Ren et al., 2017*), NB termination in the pupa (*Yang et al., 2017*; *Samuels et al., 2020b*) and is required for neuromuscular junction development and function (*McDermott et al., 2014*; *Halstead et al., 2014*; *Titlow et al., 2020*). *syp* has previously been identified in a screen for genes required for normal brain development (*Neumüller et al., 2011*), although the defect was not characterised in detail.

In light of these studies, we wanted to understand the defect caused by *syp* on brain development in more detail. We therefore examined *syp* - / - brains (eliminating Syp expression in the NB lineages) and found that in early wL3, brains were significantly enlarged compared to WT larvae at the same stage of development (p<0.0001, t-test, *Figure 4A*, *Figure 4—figure supplement 1A*). *syp* brain lobes exhibit a 23% increase in diameter (WT 206.5 μm ± 5.0, n = 10, *syp* 253.7 μm ± 11.0, n = 5), and a 35% increase in central brain (CB) volume. Significantly, a more specific RNAi knockdown of *syp* driven under the *inscuteable* promoter, which is expressed primarily in NBs and GMCs, demonstrates a similar increase in CB diameter (p=0.002, 13% larger than WT; 234 μm ± 17.0, n = 12; *Figure 4—figure supplement 1A*). Our data raises the question as to how the removal of *syp* from the neural lineages causes such a significant increase in central brain size.

We tested whether this brain overgrowth is caused by additional ectopic NBs, as has been previously described for other mutants (*Bello et al., 2006*). We used CytoCensus to accurately determine the total number of NBs in the CB of fixed *syp* knockdown verses WT wL3 brains. Our results show that wL3 brains with *syp* RNAi knockdown have no significant difference in ventral NB number compared to WT (*Figure 4B*; WT 45.6 ± 1.3, n = 22, *syp* RNAi 44.1 ± 2.1, n = 15). We conclude that a

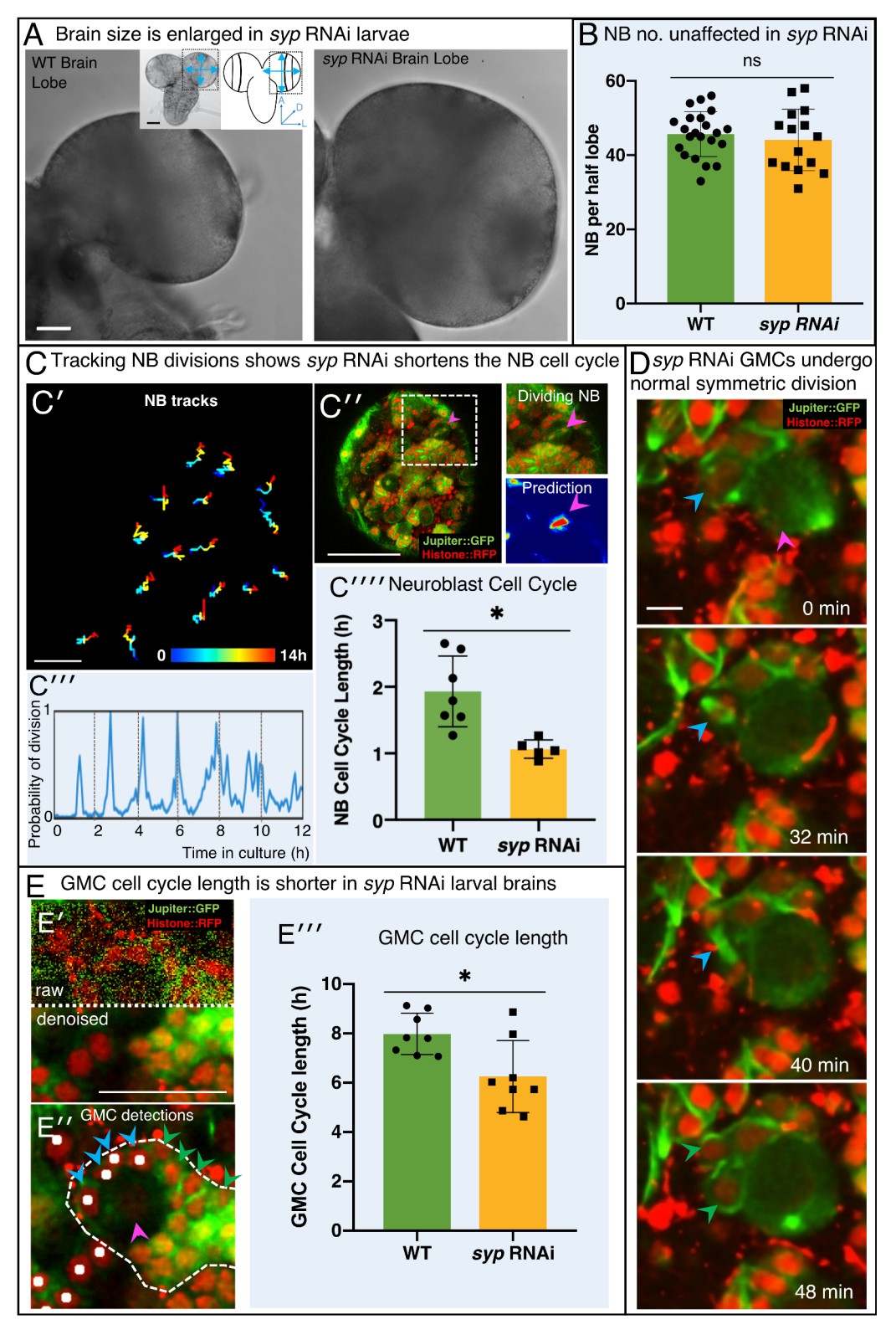

**Figure 4.** Knockdown of Syncrip protein in NBs causes larval brain enlargement. (**A**) Brightfield images of freshly isolated brains from third instar WT (OregonR) and *syp* RNAi larvae, respectively. Inserts in (**A**) show the region of the brain imaged and the measurements taken to compare brain size. (**B**) Chart comparing NB numbers showing that *syp* RNAi knockdown does not have a significant effect on NB number/brain (ns, p=0.77, t-test, WT n = 22; RNAi n = 15). NB were identified by Dpn labelling and the average count for a comparable volume of a single optic lobe CB region is shown. (**C**)

*Figure 4 continued on next page*

*Figure 4 continued*

Automated identification of NB division using CytoCensus: (C′) Tracking of NB centres, based on CytoCensus detections, over 14 h; (C″) raw image showing single timepoint from live, 3D time-lapse, confocal imaging (insert = single dividing NB, showing CytoCensus prediction of a dividing NB); (C‴) graph of division of a single tracked NB over 14 h; (C⁗) average NB (6–9 NB/brain) cell cycle length is reduced in *syp* RNAi knockdown brains (p=0.004, Welch's t-test, WT n = 7, *syp* RNAi n = 5 brains); (D) Sequence of confocal images from a typical 3D time-lapse movie showing that in *syp* RNAi brains, GMCs divide normally to produce two equal sized progeny that do not divide further. (E) Semi-automated analysis of GMC division by CytoCensus shows that GMC cell cycle length is reduced in *syp* RNAi brains. (E′) Single image plane taken from a 3D time-lapse, confocal image data set (imaged at one Z-stack/2 min). showing raw image data (top) and denoised (bottom). (E″) CytoCensus GMC detections (cyan) with a single NB (magenta), and NB niche (dotted white line), shows GMCs are detected but neurons (green) are not. (E‴) Plot of GMC cell cycle length, which is decreased in *syp* RNAi brains compared to WT (p=0.01, Welch's t-test, n = 8 GMCs from three brains). Scale bars in (A) 50 µm; (C′) 20 µm; (C″) 50 µm; (D) 5 µm; (E) 25 µm.

The online version of this article includes the following video and figure supplement(s) for figure 4:

**Figure supplement 1.** Loss of Syp causes enlarged larval brains.

**Figure 4—video 1.** Neuroblast division in live explanted larval brains under extended time-lapse imaging conditions.

https://elifesciences.org/articles/51085#fig4video1

**Figure 4—video 2.** Tracking of GMCs in a live explanted larval brain under extended time-lapse imaging conditions, collected at 2 min intervals and displayed at 5 fps.

https://elifesciences.org/articles/51085#fig4video2

change in NB number is not the underlying cause of brain enlargement observed in *syp* RNAi and hypothesise that a change in NB division rate or that of their progeny might be responsible.

## *syp* RNAi knockdown brains exhibit an increased NB division rate

To investigate whether an increase in NB division rate contributes to the brain overgrowth observed in *syp* knockdown larvae, we examined the rate of NB division in living brains using our optimised culturing and imaging methods, followed by CytoCensus detection and tracking.

First, we perform 3D NB detections using CytoCensus (as shown previously in *Figure 3A*), and we fed this input into TrackMate, a simple tracking algorithm. Without the CytoCensus detections, TrackMate spot detection performs poorly on the raw data (F1 score 0.11 ± 0.09), and tracking is all but impossible. Applying TrackMate to the proximity maps generated by CytoCensus dramatically improves TrackMate detections (F1 score 0.92 ± 0.02, *Figure 5—figure supplement 1A*). As a result, 16 out of 17 NBs were successfully and accurately tracked for over 20 h in our tests (*Figure 4C′*, *Figure 5—figure supplement 1A$^V$*).

In order to follow the NB cell cycle, we next showed CytoCensus can accurately identify individual dividing NBs in live image series, both in WT (*Figure 4C″*) and in RNAi brains (*Figure 5—figure supplement 1B*). We detected dividing NBs by training on NBs with visible spindles using CytoCensus, and used this output to create plots of division for each NB (*Figure 4C‴*, *Figure 5—figure supplement 1C–D*). Using these plots, we measured the cell cycle length of NBs in wild type and *syp* RNAi brains and found that, on average, *syp* RNAi NBs have a 1.78-fold shorter cell cycle compared to WT (p=0.02, Welch's t-test, WT N = 7, *syp* RNAi n = 5 brains; *Figure 4C⁗*). We propose that this shorter cell cycle length (i.e. an increased division rate) in the *syp* knockdown is the primary cause of its increased brain size. These results illustrate the potential of CytoCensus to analyse the patterns of cell division in a complex, dense tissue, live, in much more detail than conventional methods in fixed material.

## GMC cell cycle length is slightly decreased in *syp* RNAi brains

We also investigated GMC behaviour in the CB region of *syp* RNAi and WT larval brains, to test whether an aberrant behaviour of mutant GMCs could also contribute to a brain enlargement phenotype. Given that GMCs are morphologically indistinguishable from their immature neuronal progeny (which makes them particularly difficult to assess) we had to identify GMCs by tracking them from their birth in a NB division to their own division into two neurons. To achieve this goal required us to use high temporal resolution imaging and patch based denoising (Materials and methods), which allowed us to confirm that normal, symmetric GMC divisions occurred with the correct timing and resulted in two daughter cells (which did not regrow or divide further), both in WT and *syp* RNAi (*Figure 4D*).

Using our refined culture and imaging conditions, we trained CytoCensus to successfully detect GMCs in denoised images (*Figure 4E'-E''*) and, similarly to NBs, track them with a trackpy based script (*Allan et al., 2012*, see Materials and methods). Unlike in the case of NB tracking, GMCs do not go through repeated cycles of division, so following automated detection, for each GMC, we manually identified the birth and final division and additionally corrected any tracking errors. This semi-automated tracking allowed us to compare the cell cycle length of GMCs in multiple brains over 12 h time-lapse movies for the first time (*Figure 4E'''*). In *syp* RNAi, we find a small but significant shortening (p=0.01, Welch's t-test) of the cell cycle compared to WT (8.00 h ± 0.89, n = 8 WT; 6.25 h ± 1.45, n = 8 *syp* RNAi). However, while we conclude that GMC cell cycle length is decreased by 20%, GMCs terminally divide normally (representative example, *Figure 4D*), and we see no evidence of further divisions in the neurons. We take this to mean that no additional cells are produced by GMC or neuron division and therefore brain size is not significantly affected. We conclude that the cause of the enlarged brain size in *syp* RNAi brains is an increase in NB division rate resulting in more GMCs and their progeny than in WT.

## NB division rate is consistently heterogenous in *Drosophila* brains

Most current methods for measuring NB division rates produce an average rate for whole brains rather than providing division rates for individual NBs. It has previously been shown that NB lineages give rise to highly variable clone size (30–150 neurons for Type I neuroblasts). The origin of this diversity has primarily been attributed to patterned cell death (*Yu et al., 2013*; *Pinto-Teixeira et al., 2016*), but the importance of NB division rate in determining clone size is less well understood. Using live imaging and CytoCensus, however, we were able to quantitate the behaviour of multiple individual NBs over time within the same brain to investigate whether cell division rates are constant or variable across the population. Interestingly, we found that each NB has a constant cell cycle period (*Figure 5A*), matching observations *in vitro* (*Homem et al., 2013*). However, there is considerable variation in cell cycle length between NBs within the same brain lobe (*Figure 5A*). Given the scale of this variation, which covers more than twofold difference in rate, we expect that the regulation of NB division rate is a key factor that contributes to the observed variation in NB lineage size. By comparing the distribution of division rates in individual WT and *syp* RNAi brains, we found that *syp* knockdown NBs have a more consistent division rate in individual NBs (*Figure 5B*) and between brains (*Figure 4C'''*), which suggests a role for *syp* in the regulation of NB division rate. Future work using CytoCensus and live imaging would allow one to explicitly link individual NB division rates to atlases of neural lineages and explain the contribution of division rate to each neural lineage.

We conclude that analysing live imaging data with CytoCensus can provide biological insights into developmental processes that would be difficult to obtain by other means. However, it was important to establish the use of CytoCensus in other situations outside *Drosophila* tissues, especially in vertebrate models of development.

## Directly quantifying cell numbers enhances the analysis of zebrafish retinal organoid assembly

To test the utility of CytoCensus for the analysis of complex vertebrate tissue, we first analysed Zebrafish tissue, an outstanding model for studying development with many powerful tools, such as the Spectrum of Fates (SoFa) approach (*Almeida et al., 2014*), which marks cells from different layers of the Zebrafish retina by expression of distinct fluorescent protein labels. Previously published work by *Eldred et al. (2017)* studying eye development in artificial Zebrafish organoids, provided an excellent example of material that was previously analysed using bespoke MATLAB image analysis software that measured only the cumulative fluorescence at different radii from the organoid centre. While this was sufficient for a summary of organoid organisation, future research will require the ability to examine organoids at the single-cell level, particularly in cases where layers are formed from a mixture of cell types or cell types are defined by combinations of markers. We deployed CytoCensus to this end, without the need for bespoke image analysis, in directly locating and counting cells (*Figure 6A*).

Using CytoCensus, we trained multiple models on subsets of the raw data (*Figure 6A'*, gift from the William Harris lab), corresponding to each of the different cell types. Applying our models to the remainder of the dataset, CytoCensus was able to identify individual cells (*Figure 6A*, bottom

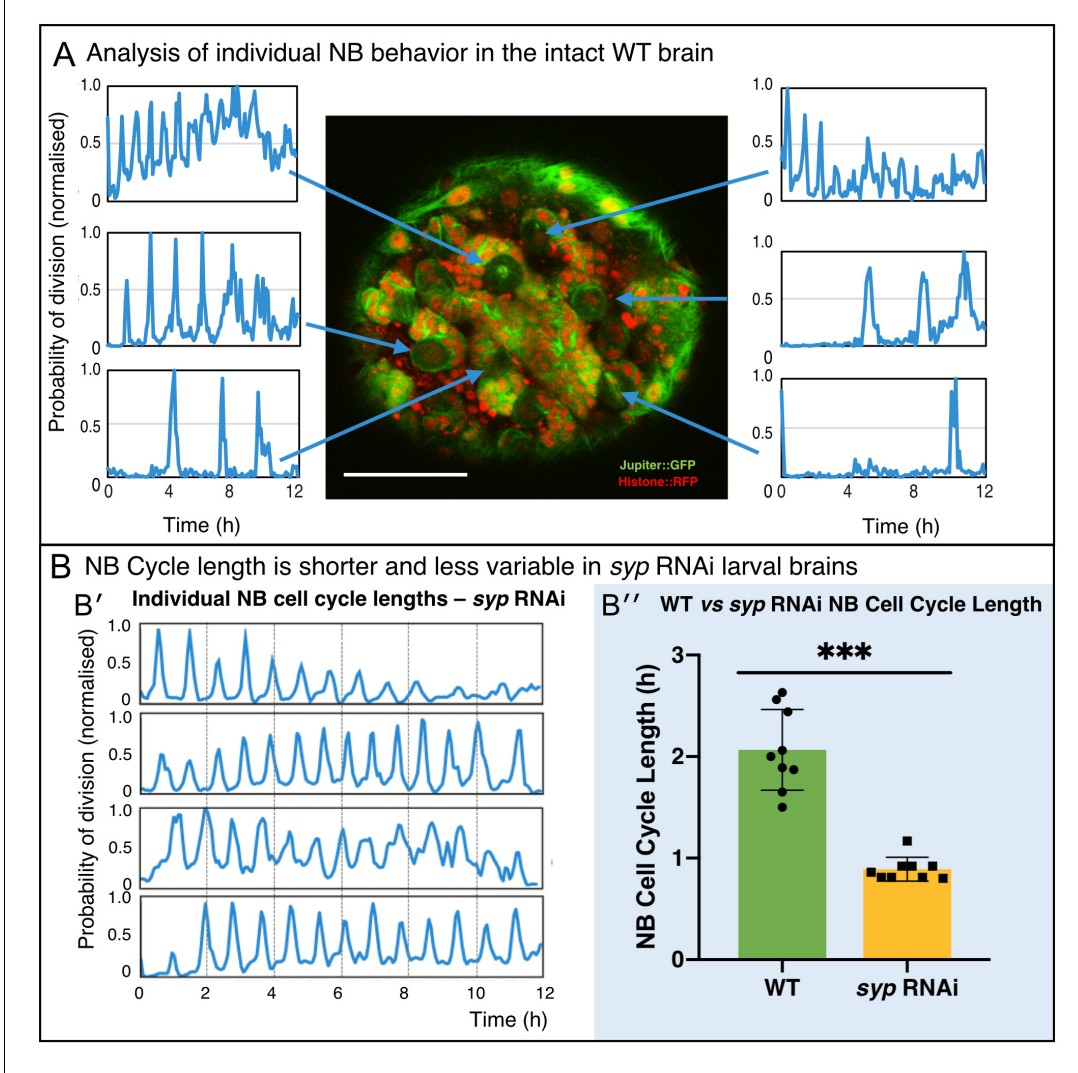

**Figure 5.** Direct analysis of NB division from time-lapse imaging of live explanted larval brains. (A) Using the proximity map output of CytoCensus, individual NBs can be followed through their cell cycle. Arrows: Individual NB locations, and the corresponding proximity map output plotted over time for that NB. (B) Comparison of WT and *syp* RNAi NB: (B′) analysis of cell cycle over time for individual NBs from a *syp* RNAi brain; (B″) comparison of cell cycle lengths for individual NB in a single WT vs *syp* RNAi brain (p=0.002, F-test, n = 9 NB). Scale bar 40 µm.
The online version of this article includes the following figure supplement(s) for figure 5:

**Figure supplement 1.** Following NB division with CytoCensus.

panels), allowing an analysis of cellular distribution that would not be possible from cumulative fluorescence measurements. We then calculated the number of cells found at different distances from the center of the organoid (Materials and methods, *Figure 6A″-A‴*). Using this approach, we reproduced the previously published analysis (*Eldred et al., 2017*), mapping the different cell distributions in the presence and absence of retinal pigment epithelium cells. We show that CytoCensus produces similar results to Figure 2 of *Eldred et al. (2017)*, but with identification of individual cells and without the need for a dedicated image analysis pipeline (*Figure 6A″-A‴*). In particular, we are able to produce an estimate of the distribution of the photoreceptor (PR) cell class, which is defined by a combination of markers (Crx::gapCFP, Ato7::gapRFP) that could not be separated from other cell types in the original analysis.

Given that the SoFa markers support the study of live organoid development, and CytoCensus can be used to identify cells based on the SoFa markers, we expect CytoCensus could easily be used to analyse live organoid development along similar lines to our *Drosophila* analysis. We

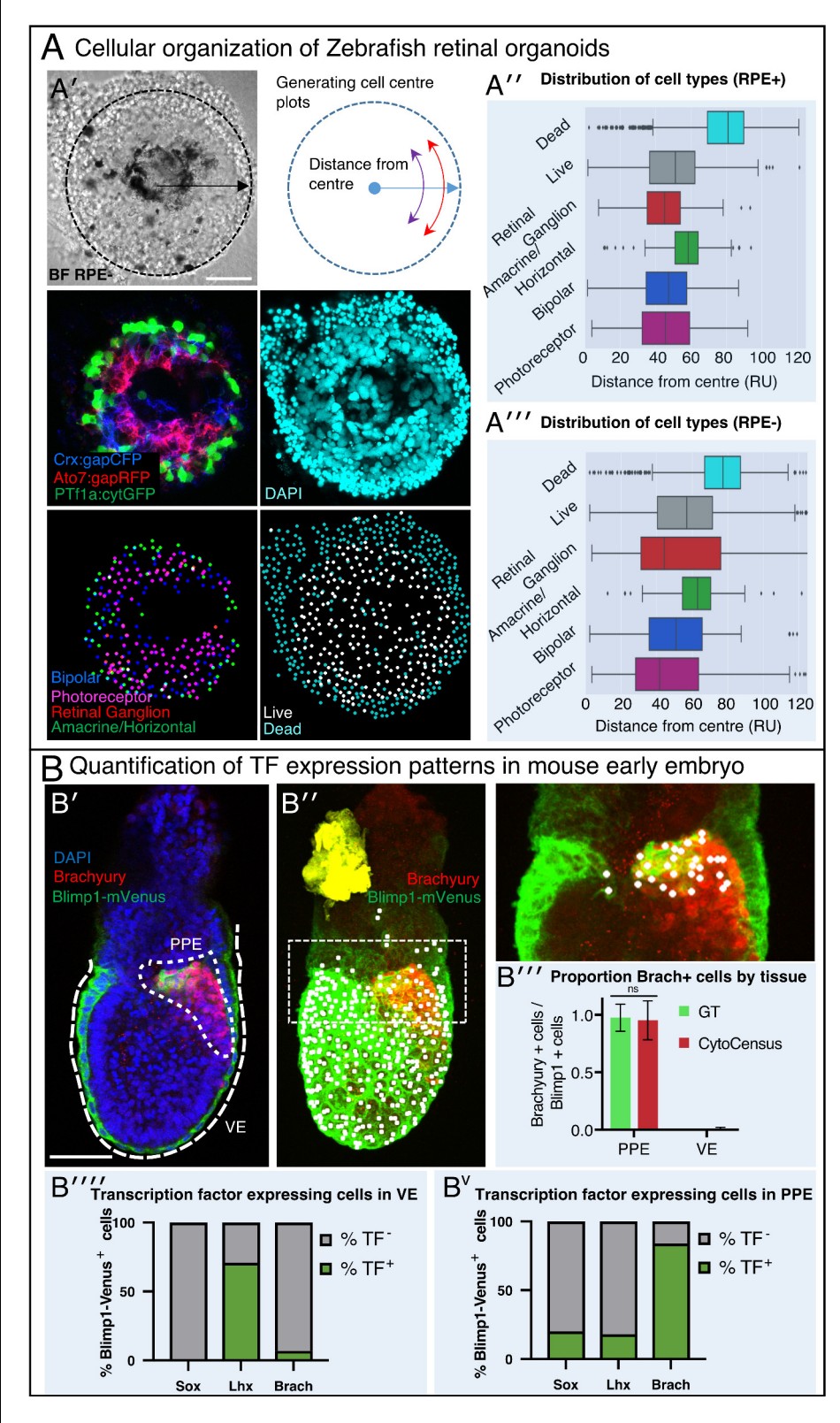

**Figure 6.** A generally applicable automated analysis tool to assess tissue development. (**A**) Automated analysis of Zebrafish retinal organoids at the single-cell level. Raw data from *Eldred et al. (2017)*. (**A′**) Top: brightfield image and diagram indicating the location of cells was defined as displacement from the organoid center. Middle: Cell fate marker expression (Crx:gapCFP; Ato7:gapRFP; PTf1a:cytGFP) and DAPI. Bottom: Cell centre identification by

*Figure 6 continued on next page*

*Figure 6 continued*

CytoCensus for the different cell types as defined by the labelling profiles (Bipolar, Photoreceptor, Retinal Ganglion, Amacrine/Horizontal, Live/Dead). (**A″-A‴**) Radial distribution of the different cell types determined from cell centre identifications by CytoCensus; the effects on organoid organisation of the presence (**A″**) or absence (**A‴**) of retinal pigment epithelium (RPE) cells is examined (ns, one-way ANOVA). RU = Radial Units, normalised to a radius of 100 (see Materials and methods) (**B**) Automated quantification of TF expressing cells in a fixed early streak stage mouse embryo (e6.5) labelled for transcription factors, Blimp1-mVenus and DAPI. (**B′**) A medial confocal section showing Brachyury in the primitive streak in the proximal posterior epiblast (PPE) and visceral endoderm (VE, highlighted cortical tracing). (**B″**) Cortical image of the same mouse embryo overlaid with total cell centre predictions by CytoCensus of Brachyury positive cells; insert to the right is a zoomed in image of the highlighted rectangle showing only cell centre predictions in a single medial plane. (**B‴**) Comparison of CytoCensus and manual Ground Truth (GT) measurements of the proportion of Brachyury positive cells from 2D planes in the VE and PPE (ns, t-test, n = 3). (**B⁗-Bv**) Proportion of transcription factor positive cells (TF) in, using CytoCensus measurements in 3D according to tissue regions (PPE and VE) defined in (**B′**). Scale bars 25 µm in (**A**); 100 µm in (**B′**).

The online version of this article includes the following figure supplement(s) for figure 6:

**Figure supplement 1.** CytoCensus detections of cells in Mouse embryos.

conclude that CytoCensus is an effective tool to investigate the distribution of cell types in the assembling retinal organoid, with the potential to analyse other complex Zebrafish tissues.

## CytoCensus facilitates rigorous quantification of TF expression patterns in mouse embryos

Mouse models are widely used to understand developmental processes in the early embryo. In such work, genetic studies have been fundamental in understanding the molecular mechanisms underlying important lineage decisions (*Piliszek et al., 2016*; *Arnold and Robertson, 2009*). However, assessment of changes in cell numbers and distribution frequently relies on manual counting and qualitative estimation of phenotypes. We tested the ability of CytoCensus to provide quantitative data on the number of transcription factor positive cells in the early post-implantation embryo for each of the transcription factors Brachyury, Lhx1 and Sox2. Using CytoCensus, we quantitated the number of cells that express each of these transcription factors in two regions of interest: the visceral endoderm (VE) and the proximal posterior epiblast (PPE), where primordial germ cells (PGCs) are specified. We also analysed the distribution of Blimp1-mVenus in membranes in both the VE and PGCs (*Ohinata et al., 2008*; *Vincent et al., 2005*; *Figure 6B′, B″*).

Using CytoCensus, we identified all Blimp1 expressing cells and mapped them to structures of interest using a 3D ROI (*Figure 6B′* marked regions). We then used CytoCensus to identify cells expressing both Blimp1 and Brachyury in the proximal posterior epiblast (PPE) (*Figure 6B″* and insert). We note that CytoCensus could be used to successfully detect cells of the VE and PGCs, despite the fact that they are frequently far from round. CytoCensus is able to detect these cells, almost as well as truly round cells, by integrating information from the nuclear and membrane markers to produce robust cell centre detections. Our analysis highlights the enrichment of Brachyury in the developing PGCs and their almost complete absence from the VE, which matches well with manual 2D quantification (*Figure 6B‴*). Repeating this analysis for the transcription factors Sox2 and Lhx1 highlights a differential expression of the transcription factors (*Figure 6B⁗-V*). These proportions match well with qualitatively reported expression patterns in the field (*Piliszek et al., 2016*). Our results demonstrate how CytoCensus can be used to produce a robust and detailed quantitation of cell type and TF expression in specific complex mouse tissues using standard markers, improving on the standard qualitative analysis.

Taking our results in their entirety, in *Drosophila*, Zebrafish and mouse, we illustrate the wide applicability of CytoCensus to transform the quantitative analysis of any complex tissue. CytoCensus makes it possible without bespoke programming to quantitate cell numbers and their divisions in complex living or fixed tissues in 3D.

## Discussion

Progress in understanding the development and function of complex tissues and organs has been limited by the lack of effective ways to image cells in their native context over extended developmentally relevant timescales. Furthermore, a major hurdle has been the difficulty of automatically analysing the resulting large 4D image series. Here, we describe our development of culturing and imaging methods that support long term high resolution imaging of the cells in intact living explanted *Drosophila* larval brains. This progress relies on optimised dissection and mounting protocols, a simplified culture medium for extending brain viability and the use of patch-based denoising algorithms to allow high resolution imaging at a tenth of the normal illumination intensity. We next describe our development of CytoCensus: a convenient and rapid image analysis software employing a supervised machine learning algorithm. CytoCensus was developed to identify neural stem cells and other cell types, both in order to quantitate their numbers and distribution and to enable analysis of the rate of division on an individual cell level, from complex 3D and 4D images of cellular landscapes. We demonstrate the general utility of CytoCensus in a variety of different tissues and organs.

To image all the cell types in an explanted brain, we used very bright generic markers of cellular morphology, which offer major advantages over specific markers of cell identity, as they are more abundant and brighter, allowing the use of low laser power to maximise viability. Markers of cell morphology can also be used in almost all mutant backgrounds in model organisms, unlike specific markers of cell identity, whose expression is often critically altered in mutant backgrounds. However, imaging cells in a tissue or organ with generic markers leads to complex images, in which it is very challenging to segment individual cells using manual or available image analysis tools. In contrast to other approaches, we demonstrate that CytoCensus allows the user to teach the program, using only a few examples, by simply clicking on the cell centres. CytoCensus outperforms, by a significant margin, the other freely available approaches that we tested, so represents a step change in the type and scale of datasets that can be effectively analysed by non-image analysis experts. Crucially, CytoCensus analysis combined with cell tracking in extensive live imaging data allows parameters such as cell cycle length to be determined for individual cells in a complex tissue, rather than conventional methods that provide snapshots or an ensemble view of average cell behaviour.

The image analysis approach we have developed depends critically on the use of 'supervision' or training regimes which are, by definition, subjective and user dependent. Supervised machine learning methods (*Luengo et al., 2017*; *Arganda-Carreras et al., 2017*; *Logan et al., 2016*; *Chittajallu et al., 2015*; *Sommer, 2011*) require the user to provide training examples by manually identifying (annotating) a variety of cells or objects of interest, often requiring laborious 'outlining' of features to achieve optimal results. Where extensive training data, appropriate hardware and expertise are available, users should consider the use of NN such as those described in *Falk et al. (2019)* because of their superior ability to make use of large amounts of training data. However, our use of a 2D 'point-and-click' interface (*Figure 2—figure supplement 1*), to simplify manual annotation, with a 3D proximity map output, and choice of fast machine learning algorithm, makes it quick and easy for a user to train and retrain the program with minimal effort. Using our approach, a user can rapidly move from initial observations to statistically significant results based upon bulk analysis of data.

We show the value of CytoCensus in three key exemplars. In *Drosophila*, we measure cell cycle lengths *ex vivo* in two key neural cell types, revealing the significant contribution of neuroblast division rate to the *syp* RNAi overgrowth phenotype. This complements a study some of the authors of this paper published while this paper was being revised (*Samuels et al., 2020a*). Samuels et al. show that Syncrip exerts its effect on NB by inhibiting Imp, which in turn regulates the stability of the mRNA of *myc* a proto-oncogene that regulates size and division. In Zebrafish organoids, we illustrate that CytoCensus is generally applicable and compatible with other cell types and live imaging markers. We show it is possible to easily characterise organoid organisation at the cellular level, including analysis of cell type which was not previously quantified (*Eldred et al., 2017*). Finally, we quantify TF expression in images of mouse embryos, illustrating how qualitative phenotypes can be straightforwardly converted into quantitative characterisations, even in epithelial tissue which differs from the typical assumptions of round cells.

A technical limitation of our 'point-and-click' strategy is that the program 'assumes' a roughly spherical cell shape. This means that cellular projections, for instance axons and dendrites of neurons, would not be identified, and other programs (e.g. Ilastik, etc.) may be more appropriate to answer specific questions that require knowledge of cell shape or extensions. However, we find that the robustness of the CytoCensus cell centres, even with irregular or extended cells can be a useful starting point for further analysis. To this end, we configured the output data from CytoCensus to be compatible with other programs, such as FIJI (ImageJ), allowing a user to benefit from the many powerful plug in extensions available to facilitate further extraction of information for defined cell populations from bulk datasets.

With the increased availability of high-throughput imaging, there is a greater unmet need for automated analysis methods. Ideally, unsupervised methods will remove the need for manual annotation of datasets, but at present, the tools required are in their infancy. In this context, methods that require minimal supervision, such as CytoCensus are desirable. Machine learning approaches, such as CytoCensus, offer the potential to analyse larger datasets, with statistically significant numbers of replicates, and in more complex situations, without the need for time-consuming comprehensive manual analysis. Easing this rate limiting step will empower researchers to make better use of their data and come to more reliable conclusions. We have demonstrated that analysis of such large live imaging datasets with CytoCensus can provide biological insights into developmental processes in *Drosophila* that would be difficult to obtain by other means, and that CytoCensus has a great potential for the characterisation of complex 4D image data from other tissues and organisms.

# Materials and methods

## Key resources table

| Reagent type (species) or resource | Designation | Source or reference | Identifiers | Additional information |
|---|---|---|---|---|
| Antibody | Guinea pig polyclonal anti-Syncrip | I.Davis Lab (*McDermott et al., 2012*) | N/A | (use 1:100) |
| Antibody | Mouse monoclonal anti-Prospero | Abcam | ab196361 | (use 1:100) |
| Antibody | Guinea pig polyclonal anti-Asense | Gift from JA Knoblich | N/A | (use 1:200) |
| Antibody | Rat monoclonal anti-Deadpan | Abcam | ab195173 | (use 1:100) |
| Antibody | Goat monoclonal anti-Mouse Alexa Fluor 488 | ThermoFischer | A-11001 | (use 1:250) |
| Antibody | Goat monoclonal anti-Guinea pig Alexa Fluor 647 | ThermoFischer | A-21450 | (use 1:250) |
| Antibody | Goat monoclonal anti-Rabbit Alexa Fluor 594 | ThermoFischer | R37117 | (use 1:250) |
| Antibody | Goat monoclonal anti-Mouse Alexa Fluor 647 | ThermoFischer | A-32728 | (use 1:250) |
| Chemical compound/drug | VECTASHIELD Antifade Mounting Medium | VECTOR Laboratories | H-1000 | N/A |
| Chemical compound/drug | Formaldehyde, 16%, methanol free, Ultra Pure | Polysciences, Inc | 18814–20 | N/A |
| Chemical compound/drug | Low melting point agarose | ThermoFischer | v2111 | N/A |
| Chemical compound/drug | Foetal Bovine Serum (FBS) | Life Technologies Ltd | 10500064 | N/A |

*Continued on next page*

*Continued*

| Reagent type (species) or resource | Designation | Source or reference | Identifiers | Additional information |
|---|---|---|---|---|
| Chemical compound/drug | Schnider's Medium | ThermoFischer | 21720024 | N/A |
| Chemical compound/drug | Bromophenol Blue | Sigma-Aldrich | 116K3528 | N/A |
| Strain (*Drosophila*) | *Drosophila* Wild-Type, Oregon-R | Bloomington | 2376 | N/A |
| Strain (*Drosophila*) | *Drosophila*: Jupiter::GFP, Histone::RFP (recombined on the third) | Ephrussi Lab | N/A | N/A |
| Strain (*Drosophila*) | *Drosophila*: AseGal4 >> UAS-MCD8-GFP | This article | N/A | N/A |
| Strain (*Drosophila*) | *Drosophila*: w11180; PBac(PB)syp e00286/TM6B | Harvard (Exelixis) | e00286 | N/A |
| Strain (*Drosophila*) | *Drosophila*: w[11180]; Df(3R)BSC124/TM6B | Bloomington | 9289 | N/A |
| Strain (*Drosophila*) | *Drosophila*:syp RNAi lines w11180; P{GD9477} v33011, v33012 | VRDC | 33011, 33012 | N/A |
| Strain (*Drosophila*) | *Drosophila*: ase-GAL4 | Gift from JA Knoblich | N/A | N/A |
| Software/algorithm | Fiji, ImageJ (V1.51d) | *Schindelin et al., 2012* | N/A | http://imagej.nih.gov/ij |
| Software/algorithm | Ilastik (V1.17) | *Sommer, 2011* | N/A | ilastik.org |
| Software/algorithm | CytoCensus | This article | N/A | github.com/hailstonem /CytoCensus |
| Software/algorithm | SoftWoRx, Resolve3D | GE Healthcare | | N/A |
| Software/algorithm | Microsoft Excel | Microsoft Cooperation | N/A | 150722 |
| Software/algorithm | OMERO V5.3.5 | *Allan et al., 2012* | N/A | openmicroscopy. org/omero/ |
| Software/algorithm | Bio-Formats | *Linkert et al., 2010* | N/A | openmicroscopy. org/bio-formats/ |
| Software/algorithm | ND-SAFIR, PRIISM | *Carlton et al., 2010* | N/A | N/A |
| Software/algorithm | Trackmate 3.8 | *Tinevez et al., 2017* | N/A | N/A |
| Other | Superfine Vannas dissecting scissors | WPI | 501778 | N/A |
| Other | MatTek (or Eppendorf) 3 cm glass-bottom Petri- dish | MatTek (or Eppendorf) | P35G-1.5–14 C | N/A |
| Other | Broad Bioimage Benchmark Collection Datasets | https://data.broadin stitute.org/bbbc/; *Svoboda et al., 2009* | BBBC024vl | N/A |
| Other | Cell Tracking Challenge datasets | celltrackingchallenge.net *Ulman et al., 2017*, *Maška et al., 2014* | N/A | N/A |

## Fly strains

Stocks were raised on standard cornmeal-agar medium at either 21°C or 25°C. To assist in determining larval age, Bromophenol Blue was added at 0.05% final concentration in cornmeal-agar medium. The following *Drosophila* fly strains were used: [Wild-Type Oregon-R]; [Jupiter::GFP;Histone::RFP (recombination on the third)]; [AseGal4 >UAS-MCD8-GFP]; [w11180;PBac(PB)syp e00286/TM6B];

[Bloomington 9289, w11180 (homozygote syp Null)]; [Df(3R)BSC124/TM6B (crossed to BL 9289 for syp Null)]; [syp RNAi lines - w11180; P{GD9477}v33011, v33012].

## Mouse embryos

Refer to *Simon et al. (2017)* for details on mouse embryo preparation.

## Fixed tissue preparation and labelling

Flies of both genders were raised as described above and larvae from second instar to pre-pupal stages collected and dissected directly into fresh 4% EM grade paraformaldehyde solution (from a 16% stock. Polysciences) in PBS with 0.3% TritonX-100 then incubated for 25 min at room temperature (RT). Following fixation, samples were washed 3 times for 15 min each in 0.3% PBST (1x PBS containing 0.3% Tween) and blocked for 1 hr at RT in Immunofluorescence blocking buffer (1% FBS prepared in 0.3% PBST). Samples were incubated with primary antibody prepared in blocking buffer for either 3 hr at RT or overnight at 4°C. Subsequently, samples were washed three times for 20 min each with 0.3% PBST followed by incubation with fluorescent labelled secondary antibodies prepared in blocking buffer for 1 hr at RT. For nuclear staining, DAPI was included in the second last wash. Samples were mounted in VECTASHIELD (Vector Laboratories) for examination. For details on the preparation and labelling of mouse embryos, refer to *Simon et al. (2017)*.

## Culture of live explanted larval brains on the microscope

Brains were dissected from 3rd instar larvae in Schneider's medium according to https://www.youtube.com/watch?v=9WlIoxxFuy0 and placed inside the wells of a pre-prepared culturing chamber (*Figure 1A*). To assemble the culturing chamber, 1% low melting point (LMP) agarose (ThermoFischer) was prepared as 1:1 v/v ratio of 1 x PBS and Schneider's medium (ThermoFisher 21720024) then pipetted onto a 3 cm Petri dish (MatTek) dish and allowed to solidify. After solidification, circular wells were cut out using a glass capillary ~2 mm diameter. To secure the material in place, a 0.5% LMP solution [1% LMP solution brain diluted 1:1 with culturing medium (BCM)] was pipetted into the wells to form a cap. Finally, the whole chamber was flooded with BCM. BCM was prepared by homogenising ten 3rd instar larvae in 200 µl of Schneider's medium and briefly centrifuge to separate from the larval carcasses. This lysate was added to 10 ml of 80% Schneider's medium, 20% Foetal Bovine Serum (GibcoTM ThermoFisher), 10 µl of 10 mg/ml insulin (Sigma). A lid is used to reduce evaporation. For GMC imaging we used a solid-agar cap (1–2% LMP agarose) placed directly on top of the brains, which we found was more consistent at holding brains against the coverslip than our earlier approach. We note that care must be taken not to flatten brains during this process, as it appears to result in a higher rate of stalled NB divisions which are likely artefacts. This approach reduced movement in brains significantly, but did not eradicate it - it seems likely remaining movement is the primarily the result of thermal drift of the microscope focus, and is well corrected using image registration.

## Imaging

Confocal, live imaging of *Drosophila* was performed using an inverted Olympus FV3000 six laser line spectral confocal fitted with high sensitivity gallium arsenide phosphide (GaAsP detectors), x30 SI 1.3 NA lens. The confocal pinhole was set to one airy unit to optimise optical sectioning with emission collection. Images were collected at $512 \times 512$ pixels using the resonant scanner (pixel size 0.207 µm) and x2 averaging). The total exposure time per Z stack (60) frames was ~20 s. For live culture and imaging, the sample was covered with a lid at 21 ± 1°C. Imaging of the GMC cell cycle required increased temporal and spatial resolution, compared to imaging NB: 2 min. time-lapse with $0.2 \times 0.2 \times 0.5$ µm resolution. Initial tests indicated that the resulting increased light dosage reduce the number of GMC divisions over time, which we consider to be a sign of phototoxicity. Therefore, we reduced the laser power by approximately a factor of 10 (to ~12µW at the objective for 488 nm, and 7µW for 561 nm), and used post-acquisition patch-based denoising NDSAFIR, by *Kervrann and Boulanger (2006)*, implemented as part of PRIISM, with adapt = 0, island = 4, zt mode and iterations = 3 by *Carlton et al. (2010)* to restore image quality. For details on imaging of mouse embryos (*Figure 6*) refer to *Simon et al. (2017)*. Details of organoid imaging can be found in *Eldred et al.*

*(2017)*. Additional live imaging was carried out on a GE Deltavision Core widefield system with a Lumencor 7-line illumination source, Cascade-II EMCCD camera and x30 SI 1.3 NA lens.

For imaging of fixed *Drosophila* material, either an Olympus FV1200 or FV1000 confocal was used with x20 0.75 dry or x60 1.4 NA. lenses. Settings were adjusted according to the labelling and were kept consistent within experiments.

For brightfield imaging (*Figure 1—figure supplement 1*; *Figure 4—figure supplement 1*; *Figure 4*), a GE Deltavision Core widefield system, Cascade-II EMCCD camera and x30 SI 1.3 NA lens was used. Measurements of brain diameters were performed by hand in OMERO. Reported measurements are the average of one measurement along the longest axis of a brain lobe (passing through the central brain and optic lobe), and another at right angles to that (typically across the medulla).

## Image analysis (summary)

All programs used for image analysis were installed on a MacBook Pro11,5; Intel Core i7 2.88 GHz;16 GB RAM. Basic image handling and processing was carried out in FIJI (ImageJ V1.51d; http://fiji.sc; *Schindelin et al., 2012*). The CytoCensus software, and additional scripts were written in Python, a detailed technical description is given below.

## CytoCensus method: application of an ensemble of decision trees framework to identify and quantitate cell classes in 4D

With the development of CytoCensus, we aim to make identification of cell types in multi-dimensional image sets as straightforward as possible. We do so by asking the user to identify cell centres for a small number of 2D image slices. Using the cell centre annotation and the estimated size of the cells, we create an initial 'proximity map' of cell centres, similar in concept to density kernel estimation approaches (*Waithe et al., 2015*; *Waithe et al., 2016*; *Fiaschi et al., 2012*; *Lempitsky and Zisserman, 2010*). In particular, the Ilastik Cell Counting module (*Berg et al., 2019*) performs 2D density counting, a related method to CytoCensus, but provides only 2D count estimates. On most biological tissues, including the 3D tissues that we have tested, 2D counting is insufficient to accurately count cells in 3D, because applying it to many slices results in repeated detections of the same cells in multiple slices. More recently, using modified density maps as an intermediate for detecting (and not just counting) cells has become more popular. Such intermediate maps are variously described as proximity maps (our preferred term), probability density maps, and Pmaps, they are well reviewed in *Höfener et al. (2018)*.

The advantage of using 'proximity map' based methods to detect and count cells has been previously documented, but these methods have not been extended to the case of 3D cell centres with 2D annotations (*Kainz et al., 2015*; *Waithe et al., 2016*). Once we have the proximity maps of the cell centres, we then apply a series of image filters, which pull out image features such as edges, and try to use these features to predict a new proximity map of cell centres. We generate this new proximity map using a machine learning algorithm known as an 'ensemble of decision trees' (*Breiman, 2001*; *Breiman et al., 1984*), which creates a series of 'decision trees' that individually predict poorly, but averaged together are a strong predictor. Once we have the new proximity maps, the location of cell centres in 3D is inferred from the 2D predictions by applying a 3D Hessian filter (see later), which enhances the detections and resolves their coordinates in the additional dimension.

The CytoCensus software has three main components in its workflow: The 2D training and evaluation algorithm, the 3D object finding algorithm and the 3D ROI drawing and interpolation algorithms. The software is written in python and includes a Graphical User Interface (GUI) written using the PyQt library. The 2D training and evaluation algorithm utilises an ensemble of random decision trees and a bank of filters which utilise the matplotlib, scipy, scikit-learn and scikit-image libraries (*Jones et al., 2001*; *Hunter, 2007*; *Pedregosa, 2011*; *van der Walt et al., 2014*). For the 2D training and evaluation algorithm, the user must provide suitable images and make annotations indicating the locations of features, objects or cells of interest within defined regions. The user annotates 2-D sections of 3D image volumes and defines rectangular regions which encapsulate areas containing cells or features of interest or just background. There are N image volumes ($I_{i=1}$, $I_2$, $I_3$,..., $I_N$) in the training set and M annotation sections where M > 1 ($A_{j=1}$, $A_2$, $A_3$,..., $A_M$). Each annotation contains a region of interest ($R_{j=1}$, $R_2$, $R_3$,..., $R_M$) and also a set of corresponding points ($P_{j=1}$, $P_2$, $P_3$,...,

$P_M$) with one or more dot/points $Pj = \{pt_{c=1}, pt_2, \ldots, pt_C\}$ or no points if the region only contains background. It is worth noting that providing sufficient area that does not contain cells is important for minimising false positives. As the model is designed to distinguish cells from the background it maybe appropriate to annotate regions as empty so as to acclimatise the model to the background. The points and regions are supplied by the user as they label the centroid locations of cells or objects within the image plane of interest. For each annotation, we produce a centre-of-mass representation ($F_{j=1}, F_2, F_3 \ldots, F_M$) which for each pixel (p) is defined as the maximum value of all the Gaussian kernels (N) centred on dot annotations which overlap this pixel:

$$\forall_p \in R_j, F_j^0(p) = max[\mathcal{N}(p; pt, \sigma^2 \boldsymbol{I_{2\times2}}), \forall_{pt} \in P_j]$$

and $\sigma = [\sigma_x, \sigma_y]$. The kernel is isotropic ($\sigma_x = \sigma_y$) as long the features or cells of interest are roughly spherical. For this application, we recommend choosing a sigma which is smaller than the radius of the cells or features. The Gaussian will weight pixels in the centre of cells more highly than those towards the edges or in the background. Finding the maximum pixel, rather than summing pixels amongst all the overlapping Gaussians, ensures that pixels at the edges of objects, but overlapping, are not more highly weighted than pixels that are central and represent the centre of the cells, allowing better separation of close objects.

For each pixel in the annotation region, we calculate a feature vector which describes the corresponding image pixels. Each descriptor of the feature vector is created through processing of the input image or volume with one of a bank of filters which includes: Gaussian, Gaussian Gradient Magnitude, Laplacian of Gaussian, and the minimum and maximum eigenvalues of curvature (*Fiaschi et al., 2012*). These filter kernels are applied at multiple scales (sigma = 1,2,4,8,16) to aggregate data from the surrounding pixels into the feature descriptor at that specific pixel. This scale range was appropriate for all the cases used in this study and were not changed.

Once training data has been supplied by the user and the pixel features calculated, an ensemble of random decision trees is used to learn the association between input pixels and the 'proximity map' centre-of-mass representation (*Geurts et al., 2006*). The decision tree framework was parameterised as follows: the data was sampled at a rate of 1/5 from the input regions, with 30 trees generated during training, with a depth of 10 levels and a minimum split condition of 20 samples for each node. At each node n/3 features were considered. Once trained, the decision tree framework can be applied to unseen images (without user annotation), requiring only input features to be calculated. Evaluation of images produces a centre-of-mass representation of where the cell centres are located, highly similar to the representation used during training.

The 3D object finding algorithm is applied to the output images of the random decision tree framework and involves multiple steps. First, the output images of the decision framework are rearranged into a 3D volume, this provides a representation of the proximity of cell centres in 3D. To facilitate the object identification, we next apply a determinant of Hessian blob detector which smooths our signal and also enhances objects of a specific size (*Lindeberg, 1994*). Using this filter greatly simplifies our cell identification procedure, although some idea of the size of the object is required, $h = [h_x, h_y, h_z]$ (where $h_x = h_y$ if the object is spherical in two dimensions and $h_x = h_y = h_z$ if the object is spherical in three dimensions). Finally, a 3D maxima finding algorithm is used to identify the centroid locations of the enhanced objects present in the Hessian filtered image (*Gao and Kilfoil, 2009*). A simple threshold is used to set the sensitivity to detected maxima.

To allow for selective application of the 3D counting algorithm in distinct regions of a tissue (for instance the primitive streak in *Figure 6*), and over time, a novel Region Of Interest (ROI) interpolation algorithm was introduced. The user defines a ROI by clicking points around an area of interest in a single image (e.g. top of tissue region). The user then defines another region either at the other end of the object (e.g. bottom of tissue region) or partially through the region. The algorithm can then interpolate between these user-defined ROI to create a ROI for each frame in the image-volume. The User can then repeat this process in subsequent time-frames, and the algorithm will interpolate the ROI between frames creating a smooth transition which can be tweaked through the addition of further user defined regions to smoothly follow a 3D region of the tissue over time. The interpolation is performed using bilinear interpolation of points sampled uniformly along the user defined ROI. Objects or cells with a centroid position within the tissue region can then be filtered

from the image volume allowing for selective counting and location of cells over-time within the defined region.

## Algorithm validation and comparison datasets

Validation of algorithm performance is critical in developing an effective tool. We used both real and artificial data sets to assess performance. For our baseline performance tests of CytoCensus, we quantified the number and location of NBs identified in five time-points from a movie sequence. We then compared the output from CytoCensus to that of other algorithms applied to the same test dataset. In each case, we attempted to optimise the parameters used, based, whenever possible, on the published information on two time-points that were not used for final evaluation. For TrackMate detections we used detection diameter of 20 pixels, and five conservatively set filters (standard deviation, max intensity, mean intensity, contrast). For RACE, we used the histone (nuclear) marker for the seeds, and set parameters at default except for (Max segmentation area 100, min 3D area 20, Closing 2–6, Threshold 0.0002, H-maxima 30). For Ilastik, we used features from (sigma 0.3, 1.0, 3.5, 10) for color, edge and intensity. For FIJI WEKA, we used default parameters, followed by filtering to remove small objects. For CytoCensus we used default parameters, except for object size, which was set at radius 8.

To carry out a direct comparison of algorithm performance, we used artificial 'neutral challenge' datasets of highly clustered (75%) synthetic cells, in 3D, with a low signal-to-noise ratio (SNR), obtained from the Broad Bioimage Benchmark Collection (image set BBBC024vl: *Svoboda et al., 2009*). This data has an absolute ground truth and provides a good measure of comparison for the performance of different algorithms. As CytoCensus is designed to identify cell centres, the ground truth for the Neutral Challenge data and Ilastik segmentation results were adapted to report estimated cell centres (centroids) rather than segmented boundaries, before carrying out comparison of algorithm performance. In both cases, Ilastik and CytoCensus were trained on a single image, parameters optimised over five image, and performance evaluated over the remaining (25) images. For the neuroblast dataset, detections within 1/2 a neuroblast radius were considered correct. For the BBBC dataset, detections were considered correct if they were within a stricter four pixel radius (~1/8 cell size). At the more generous 1/2 a cell size, CytoCensus reached perfect precision and recall, but Ilastik's F1-score remained below 0.4 primarily due to the problem of merged cells.

## F1-score: Objective analysis of algorithm performance

To quantitate how CytoCensus analysis of complex multidimensional image data is performing and to compare performance to other freely available programs, we made use of the weighted mean of the true and false positive identification rates, known as the F1-score (maximum value 1.0; *Chinchor, 1992*; *Figure 3* and *Figure 3—figure supplement 1*, *Table 1* and *Table 2*). F1-score is intuitively similar to accuracy: strictly it is the harmonic mean of the fraction of detections that were correction (precision) and the fraction of cells correctly identified (recall). This metric, therefore, takes into account both false positives and false negatives. Such analysis is particularly useful in parameter determination for optimum algorithm performance in applying to experimental data sets and in optimising algorithms for systematic comparison on test data.

## Cell tracking challenge segmentation benchmark

To apply CytoCensus to the Cell Tracking Challenge Segmentation Benchmark datasets we downsampled all *Figures 2–4x* before processing to increase speed of processing, although better results might be achieved without downsampling. We trained CytoCensus on 2–4 frames selected (from the training set) to capture the imaging variation between datasets and selected appropriate object sizes for each of the images. Following object detection, we create a crude segmentation by creating a sphere of corresponding size around each detected object, and refined this segmentation using a small number of iterations (2-6) of MorphACWE (*Marquez-Neila et al., 2013*), an active contour segmentation method, followed by marker-based watershed (*Meyer and Beucher, 1990*) from the CytoCensus centres in order to separate detected objects. This approach is highly dependent on the quality of the detected centres to determine the number of objects and assumes cells are approximately round, so is not well suited to tasks with extended, misshapen objects. The code for

the cell tracking challenge, including dataset specific parameters, is available at github.com/hailsto-nem/CTC_CytoCensus.

## Parameter optimisation for best performance

The algorithm underlying CytoCensus requires a range of parameters to be set, described in detail above. To simplify usage, we set most most parameters to reasonable default values. Parameters, such as the threshold setting, were assessed systematically, using the objective measures of performance described above with various data sets. This approach helped us to define which parameters should be fixed and which need to be user-modified. Details of user defined parameters and how to assess and set appropriate values are documented in the User Manual. The value of Sigma, which sets the scale of the object of interest, is particularly critical for detection. Optimum Sigma value was assessed systematically and the optimum found to be slightly smaller than the size of the cell type of interest (*Figure 3—figure supplement 1A''*), this parameter was subsequently defined as 'Object Size' (in pixels).

The level of training required is also important in supervised machine-learning approaches. We assessed the level of training required to achieve good detection of cell types of interest with different datasets (*Figure 3—figure supplement 1A'*). In all cases tested, successful identification of NBs or progeny required minimal user training (of the order of tens of examples on only a few image planes) and increasing training gave only marginal improvement (*Figure 3—figure supplement 1A'*). This is advantageous for quickly annotating datasets, but may limit the flexibility for learning in particularly difficult and complex cases.

## Cell identification

To facilitate development of CytoCensus for the study of the larval brain, we initially tested performance on multichannel 3D image datasets of fixed material where NBs and GMC's were defined by specific immunostaining (for Ase and Dpn; *Neumüller et al., 2011*; *Bayraktar et al., 2010*; *Boone and Doe, 2008*; *Figure 3—figure supplement 1A,B*). We confirmed that, given these ideal markers, NB's and GMC's could be identified. After this initial development, we extended the application of CytoCensus to our live cell imaging data with generic cytological markers. To show that cell types could be recognised correctly with the generic marker combination used for live imaging, we carried out specific immunostaining on Jupiter::GFP/Histone::RFP expressing larval brains. Manual and automated annotations, first based upon generic labels alone, were scored against identification using the specific labelling (*Figure 3—figure supplement 1C*). The results of this assessment show that, for NB (96% ± 4 Dpn positive, n = 12, three repeats) and their progeny (92% ± 2 Pros positive, n = 189, three repeats), our imaging of the generic labels supports identification of NB and progeny by CytoCensus after training (*Figure 3—figure supplement 1D*). Using a combination of these different datasets, we refined the workflow of CytoCensus and optimised the key parameters of the algorithm.

## Image pre-processing and downstream data analysis

CytoCensus outputs proximity map images and object centre XYZ co-ordinates, both of which may be used as the starting points for further data analysis pipelines, for example as seeds in watershed segmentation to determine cell areas, volumes or quantitate fluorescence intensity. CytoCensus outputs (tif files) which can easily be passed to Fiji (ImageJ) or custom analysis scripts. In this study, we illustrate several examples of extending data analysis using outputs from CytoCensus.

In our analysis of NB division rates, we generate plots of cell division over time for individual NB from the map output and NB centre coordinates (*Figure 5A,B* and *Figure 5—figure supplement 1*). Here, we used a custom trackpy based python script to track individual NBs over time, however, similar results can be achieved simply, using ImageJ ROI tools, or at scale using TrackMate (*Figure 5—figure supplement 1A*). For each of these tracks, we follow the changes in the dividing NB proximity map that correspond to division. Robustness of the division plots is further improved by subtracting a moving average over about 20 image frames, which removes spatial differences in the background of the probability density maps. For analysis of long imaging series, such as multiple NB divisions, it is important to follow individual cells over time. For such analysis, it is necessary to spatially register the individual Z stacks across time, to correct for image drift due to movements in culture, prior to

applying CytoCensus. This was achieved with an ITK based python script (http://www.simpleitk.org/SimpleITK/resources/software.html), but similar results can be achieved using the Correct 3D drift plugin in ImageJ (it is crucial to use a high noise threshold, ignoring low value pixels, as this approach is sensitive to noise). Similar approaches were employed in our analysis of GMC cell cycle length (*Figure 4D*). In the challenging case of GMCs, it was necessary to explicitly track cells as they moved significantly during development of the brain. To achieve this, estimated GMC centres were passed to a trackpy (*Allan et al., 2016*), based custom python script to perform linkage analysis and track each individual GMC over time. Similar results may be achieved using TrackMate in FIJI (*Tinevez et al., 2017*). Python scripts for further analysis are available on the CytoCensus Github page.

## Quantification and statistical comparison

Mutant comparisons were performed using an appropriate test in GraphPad Prism (see Figure legends for specific tests), typically a Student's T test, following Shapiro-Wilk test to test normal distribution of the data. Appropriate tests were selected depending on data type and normality: (*Figure 1*: one-way ANOVA to determine differences between any time-point, *Figure 3A*: one-way RM-ANOVA with post-hoc t-tests to determine difference between CytoCensus performance and the other programs, *Figure 3B*: Welch's t-test (unequal variance) to determine if there is difference between CytoCensus and Ilastik performance *Figure 4*: t-test or Welch's t-test (following F-test for variance) to determine differences in NB number or cell cycle lengths respectively. *Figure 5*: F-test to determine difference in variance between *syp* RNAi and WT. *Figure 6*: t-test to determine difference between CytoCensus and manual annotation).

A p-value of <0.05 was considered significant. Numbers of replicates typically refer to the number of independent brains and are detailed in the figure legends and main text. Measurements of cell cycle lengths/division rates are the average of 2–7 measurements (NB that only divided once were excluded) from 5 to 10 NB per brain (i.e. each NB contributes one measurement). NB Numbers were limited by the number of visible NB within the imaged region. For live imaging, sample sizes were as large as reasonably practical given the capture and processing time. For the purposes of *Figures 1* and *4*, independent brains were considered biological replicates, and NB considered technical replicates. For the purpose of comparing variation in division rates, in *Figure 5*, each NB is considered as a biological replicate, and each measurement of cell cycle length is a technical replicate. Unless otherwise stated, error bars shown are standard deviation.

## Acknowledgements

We are grateful to: Ivo A Telley (Instituto Gulbenkian de Ciência) for fly stocks; the Harris and Robertson labs for sharing their imaging data; Jordan Raff and Russel Hamilton for their insightful comments on the results; David Ish-Horowicz and Alfredo Castello for discussions and critical reading of the manuscript. We thank Tomek Dobrzycki for his contribution to the initial characterisation of the syp mutant phenotype. Thanks to Andrew Jefferson and MICRON (http://micronoxford.com, supported by a Wellcome Strategic Awards 091911/B/10/Z and 107457/Z/15/Z to ID) for access to equipment and assistance with imaging techniques.

## Additional information

### Competing interests

Elizabeth Robertson: Reviewing Editor, eLife. The other authors declare that no competing interests exist.

### Funding

| Funder | Grant reference number | Author |
| --- | --- | --- |
| Engineering and Physical Sciences Research Council | EP/L016052/1 | Martin Hailstone |
| Medical Research Council | EP/L016052/1 | Martin Hailstone |

| Biotechnology and Biological Sciences Research Council | EP/L016052/1 | Martin Hailstone |
|---|---|---|
| Medical Research Council | MR/K01577X/1 | Dominic Waithe Ilan Davis |
| Engineering and Physical Sciences Research Council | MR/K01577X/1 | Dominic Waithe Ilan Davis |
| Biotechnology and Biological Sciences Research Council | MR/K01577X/1 | Dominic Waithe Ilan Davis |
| Medical Research Council | MC_UU_12010 | Dominic Waithe |
| Medical Research Council | MR/S005382/1a | Dominic Waithe |
| Medical Research Council | MC_UU_12009 | Dominic Waithe |
| Medical Research Council | G0902418 | Dominic Waithe |
| Medical Research Council | MC_UU_12025 | Dominic Waithe |
| Oxford EPA Cephalosporin Graduate Fund | | Dominic Waithe |
| Wellcome | 105363/Z/14/Z | Tamsin J Samuels |
| Oxford University Press | Clarendon Fellowship | Lu Yang |
| Israel Science Foundation | 1096/13 | Yoav Arava |
| Wellcome | 214175/Z/18/Z | Ita Costello Elizabeth Robertson |
| Wellcome | 107457/Z/15/Z | Dominic Waithe Richard M Parton Ilan Davis |
| Wellcome | 081858 | Richard M Parton Ilan Davis |
| Wellcome | 091911/B/10/Z | Richard M Parton Ilan Davis |
| Wellcome | 096144/Z/17/Z | Richard M Parton Ilan Davis |
| Wellcome | 209412/Z/17/Z | Richard M Parton Ilan Davis |

The funders had no role in study design, data collection and interpretation, or the decision to submit the work for publication.

## Author contributions

Martin Hailstone, Conceptualization, Data curation, Software, Formal analysis, Validation, Investigation, Visualization, Methodology, Writing - original draft, Writing - review and editing; Dominic Waithe, Conceptualization, Resources, Data curation, Software, Supervision; Tamsin J Samuels, Resources, Data curation, Investigation; Lu Yang, Conceptualization, Supervision, Investigation, Methodology; Ita Costello, Resources, Visualization; Yoav Arava, Conceptualization, Methodology; Elizabeth Robertson, Resources, Supervision, Funding acquisition; Richard M Parton, Conceptualization, Supervision, Funding acquisition, Investigation, Visualization, Methodology, Project administration, Writing - review and editing, Writing - original draft; Ilan Davis, Conceptualization, Supervision, Funding acquisition, Project administration, Writing - review and editing

## Author ORCIDs

Martin Hailstone https://orcid.org/0000-0001-9326-3827
Dominic Waithe https://orcid.org/0000-0003-2685-4226
Tamsin J Samuels https://orcid.org/0000-0003-4670-1139
Yoav Arava https://orcid.org/0000-0002-2562-9409
Elizabeth Robertson https://orcid.org/0000-0001-6562-0225

Richard M Parton  https://orcid.org/0000-0002-2152-4271
Ilan Davis  https://orcid.org/0000-0002-5385-3053

## Decision letter and Author response

Decision letter https://doi.org/10.7554/eLife.51085.sa1
Author response https://doi.org/10.7554/eLife.51085.sa2

## Additional files

### Supplementary files

• Transparent reporting form

### Data availability

The following freely available image analysis tools were used: Fiji, ImageJ V1.51d (http://fiji.sc, Schindelin et al., 2012); Ilastik (V1.17) (http://ilastik.org; Logan et al., 2016; Sommer, 2011). The CytoCensus software is available open-source under GPLv3 and can be installed as a stand-alone program: full install available http://github.com/hailstonem/CytoCensus. Image data was archived in OMERO V5.3.5 (Allan et al., 2012; Linkert et al., 2010); image conversions were carried out using the BioFormats plugin in Fiji (Linkert et al., 2010); http://imagej.net/Bio-Formats). Code to run CytoCensus on the Cell Segmentation Benchmark can be found at http://github.com/hailstonem/CTC_CytoCensus.

The following previously published datasets were used:

| Author(s) | Year | Dataset title | Dataset URL | Database and Identifier |
|---|---|---|---|---|
| David S, Michal K, Stanislav S | 2009 | Generation of Digital Phantoms of Cell Nuclei and Simulation of Image Formation in 3D Image Cytometry | https://data.broadinstitute.org/bbbc/BBBC024/ | Broad Bioimage Benchmark Collection, BBBC024vl |
| Maška M, Ulman V, Svoboda D, Matula P, Ederra C, Urbiola A, España T, Venkatesan S, Balak DM, Karas P | 2014 | A benchmark for comparison of cell tracking algorithms | http://celltrackingchallenge.net/3d-datasets/ | Cell Tracking Challenge, 3d-datasets |

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
