## [Decision Letter]

**Acceptance summary:**

Both reviewers feel that the revisions are acceptable and feel that the analysis tool will be helpful to future investigations.

**Decision letter after peer review:**

Thank you for submitting your article "CytoCensus: mapping cell identity and division in tissues and organs using machine learning" for consideration by *eLife*. Your article has been reviewed by two peer reviewers, and the evaluation has been overseen by a Reviewing Editor and Richard White as the Senior Editor. The reviewers have opted to remain anonymous.

The reviewers have discussed the reviews with one another and the Reviewing Editor has drafted this decision to help you prepare a revised submission.

Summary:

In this manuscript, Hailstone et al. present CytoCensus, a supervised machine learning-based application for image analysis, which can be used to identify cells with specific attributes in 3-dimensional objects over time. Hailstone et al. first describe a new *ex vivo* brain culture technique they developed for time-lapse imaging and then use this technique to generate time-lapse data to test the performance of CytoCensus in identifying neuroblasts, compared to other available tools, namely Ilastik, Fiji-WEKA, RACE, and TrackMate. They found that CytoCensus outperforms all of them both in their data and in a neutral challenge dataset consisting of highly clustered synthetic cells. They then use CytoCensus to compare wild-type fly brains with *syp*RNAi brains, which are larger. They first exclude an increase in the number of neuroblasts and then find that the difference between the two brains is the division rate of the neuroblasts, which has as a result the production of more neurons. They also find a reduction in the cell cycle time of GMCs, which however does not contribute to the increase in size as the GMCs divide only once in both brains. They then use the proximity map output of CytoCensus to track individual NBs and verify the reduction in cell cycle length. Finally, they show that CytoCensus can also be used for developmental studies in other systems as well, testing it in zebrafish retinoids and early mouse embryos.

Essential revisions:

Both reviewers found the method useful but reviewer 2 felt that more unbiased and rigorous testing compared to other approaches were needed for benchmarking. In addition, this reviewer had questions about the novelty of the technique. After discussion, both reviewers were in agreement that improvements to the manuscript were needed.

1) The authors should discuss the novelty of their approach with respect to similar methods (Swiderska-Chadaj et al., 2018; Liang et al., 2019; Höfener et al., 2018).

2) The authors choose to use a Random Forest approach instead of using deep learning, which seems to be the best performing method for similar tasks and justify this choice by the hardware requirements of deep learning. However, it would be important to understand how large the drop in accuracy would be when compared to deep learning.

3) The authors should provide a convincing benchmarking: a) When comparing to image segmentation methods, the authors should provide a state-of-the art postprocessing scheme to make a fair comparison.

b) The authors should compare to the cell counting module available in Ilastik: https://www.ilastik.org/documentation/counting/counting.html

c) The authors should use challenge data to benchmark their results: Data Science Bowl challenge on nuclear segmentation (https://www.kaggle.com/c/data-science-bowl-2018). This challenge is on nuclear segmentation, but the authors could compare their method to the methods of a leading participant with respect to detection accuracy only.

4) The authors chose not to explain their method in the main text, which I find disturbing given that it is the major subject. The authors should describe their method in the main text in sufficient detail.

5) In the manuscript, there is a confusion between tools and methods. A software tool can be the concrete implementation of one method, but in most cases, a single tool (such as Ilastik) contains a range of methods. The authors should refer to both tool and method, in particular when they discuss benchmarking. In particular, they compare detection with segmentation methods, which is not rigorous.

[Editors' note: further revisions were suggested prior to acceptance, as described below.]

Thank you for resubmitting your work entitled "CytoCensus, mapping cell identity and division in tissues and organs using machine learning" for further consideration by *eLife*. Your revised article has been evaluated by Richard White as the Senior Editor, a Reviewing Editor and two peer reviewers.

The manuscript has been improved but there are some remaining issues from reviewer 2 that need to be addressed before acceptance, as outlined below. In particular the reviewer feels that the comparison with IIlastik needs to be fair and suggests an approach to represent this comparison more clearly.

Reviewer #2:

The authors have substantially improved the article and added a number of benchmarking results which make the manuscript stronger and the presented tool more convincing. I still believe that the result shown in Figure 3 needs to be replaced, as detailed below.

While the level of novelty remains relatively low, at least from a methodological point of view, I agree that the tool can be of much use to the community, thanks to its capacity of dealing with 3D data.

Detailed comments:

1) As mentioned in my previous review, I still do not agree with the way the results are presented in Figure 3: the authors compare their results to Ilastik. Concretely, they use Ilastik to segment cells (via pixel-classification). In a second step, they calculate the cell centers from this segmentation result as the centers of the connected components. An example of this is shown in Figure 3B' (2 quotes, upper right) and the corresponding statistics in Figure 3B' (4 quotes, bottom left). The problem with this benchmark is that this is not a reasonable approach for cell center detection. For this reason, I feel that the comparison is misleading and should not be published in this way. Normally, one would always try to apply at least some simple postprocessing for object splitting, prior to calculating the centers. The authors did this additional analysis (cited in the text) in the revised version of the manuscript, and achieved 0.88 ± 0.09 vs. 0.98 ± 0.05 with CytoCensus. This is the proper result to be reported. The difference is less striking, but it is also more realistic and will ultimately convince the readers more than the version that is currently in the manuscript. I therefore require that the Figure 3B' (4 quotes, bottom left) and Figure 3B' (2 quotes, upper right) are replaced by the corresponding figures from this more realistic scenario. The comparison to Ilastik pixel classification without post-processing should be removed.

2) Regarding the results presented in Figure 3—figure supplement 2, the authors should also provide the rank of their method (ranked x out of N participants).

---

## [Author Response]

Essential revisions:1) The authors should discuss the novelty of their approach with respect to similar methods (Swiderska-Chadaj et al., 2018; Liang et al., 2019; Höfener et al., 2018).

We have added the references and discussion as requested. We have modified the text to refer to these methods in the Introduction, making clear they share the idea of a proximity map, but also making the distinction that our method is targeted at 3D microscopy images.

In the Introduction:

“our training workflow outputs a “proximity” map, similar to those described in (Fiaschi et al., 2012, Swiderska-Chadaj et al., 2018, Liang et al., 2019, Höfener, 2018). These approaches all focus on 2D proximity maps, while CytoCensus utilises proximity maps in 3D” (which is more accurate as explained below).

We elaborate on this further in Materials and methods:

“Using the cell centre annotation and the estimated size of the cells, we create an initial ‘proximity map’ of cell centres, similar in concept to density kernel estimation approaches (Waithe, et al., 2016, Fiaschi et al., 2012, Lempitsky and Zisserman, 2010). […] Such intermediate maps are variously described as proximity maps (our preferred term), probability density maps, and Pmaps, they are well reviewed in (Höfener, 2018).”

2) The authors choose to use a Random Forest approach instead of using deep learning, which seems to be the best performing method for similar tasks and justify this choice by the hardware requirements of deep learning. However, it would be important to understand how large the drop in accuracy would be when compared to deep learning.

To address the comment, we now provide additional comparisons to the Cell Tracking Challenge Segmentation Benchmark (explained in further detail below), which allows direct comparison to a range of methods including deep learning.

We have added additional discussion to make our choice of random forest algorithms more explicit in the text. In essence, Random forest allows the user to train the tool very efficiently using a small amount of data. The authors of Ilastik make a very similar point in their recent paper in Nature Methods (Berg et al., 2019).

In Motivation and design:

“For the machine learning, we choose a variation of Random Forests with pre-calculated image features, which allows for much faster training compared to neural networks on typical computers, and with a fraction of the user annotation. A similar approach is taken by the image analysis software Ilastik (Berg et al., 2019).”

And in the Discussion:

“Where extensive training data, appropriate hardware and expertise are available, users should consider the use of NN such as those described in Falk et al., 2019 because of their superior ability to make use of large amounts of training data. However, our use of a 2D “point-and-click” interface (Figure 2—figure supplement 1), to simplify manual annotation, with a 3D proximity map output, and choice of fast machine learning algorithm, makes it quick and easy for a user to train and retrain the program with minimal effort.”

3) The authors should provide a convincing benchmarking: a) When comparing to image segmentation methods, the authors should provide a state-of-the art postprocessing scheme to make a fair comparison.

We feel the best way to address this comment is by providing an additional comparison to the Cell Tracking Challenge Segmentation Benchmark. This approach allows us to compare the output of CytoCensus to a variety of state-of-the-art methods, including tools using cutting-edge post-processing schemes (more details in Response 3c).

b) The authors should compare to the cell counting module available in Ilastik: https://www.ilastik.org/documentation/counting/counting.html

The Ilastik Cell Counting module uses a similar underlying method to CytoCensus, but provides only 2D count estimates. On most biological tissues, including the 3D tissues that we have tested, a 2D approach is inaccurate, as it results in repeated detections of the same cells in multiple slices. We have addressed this reviewer comment by adding an explicit explanation of this issue in the Results section of the manuscript:

“Ilastik Density Counting (which takes a related approach to CytoCensus) was promising to count NB in 2D, but is not designed to work in 3D nor to detect cell centres (Berg et al., 2019).”

And in the Materials and methods:

“In particular the Ilastik Cell Counting module (Berg et al., 2019) performs 2D density counting, a related method to CytoCensus, but provides only 2D count estimates.”

c) The authors should use challenge data to benchmark their results: Data Science Bowl challenge on nuclear segmentation (https://www.kaggle.com/c/data-science-bowl-2018). This challenge is on nuclear segmentation, but the authors could compare their method to the methods of a leading participant with respect to detection accuracy only.

The dataset and algorithms suggested by the reviewer are a suitable comparison for 2D methods, but given that 3D detection is a key novelty of our approach, we feel this comparison is not appropriate. We have revised the manuscript to emphasise that CytoCensus uses simple 2D annotations to extract 3D cell centres (see Responses 1, 2, 3b).

Nevertheless, we address the comment by providing an additional comparison to 3D datasets available in the Cell Tracking Challenge (CTC) Segmentation Benchmark (Figure 3—figure supplement 2B, see also Results), which allows us to directly compare detection accuracy against competitive state-of-the-art approaches. We have added the results of the benchmark to Figure 3—figure supplement 2B. This benchmarking shows that CytoCensus performs well on difficult low SNR 3D challenges when compared to the top 3 methods listed by the CTC, which includes neural networks. Based on these results, we have added the following to the Results section:

*“*For a more general comparison to other detection and segmentation methods, we applied CytoCensus to 3D data from the Cell Tracking Challenge Segmentation Benchmark (Ulman et al., 2017, Maska et al., 2014). […] In general CytoCensus performs competitively on the tested datasets, leveraging a small amount of training to achieve good results without needing dataset specific algorithms for denoising or object separation.”

We have also added a new description to the Materials and methods section:

“To apply CytoCensus to the Cell Tracking Challenge Segmentation Benchmark datasets we downsampled all images 2-4x before processing to increase speed of processing, although better results might be achieved without downsampling. […] The code for the cell tracking challenge, including parameters is available at github.com/hailstonem/CTC_CytoCensus.”

4) The authors chose not to explain their method in the main text, which I find disturbing given that it is the major subject. The authors should describe their method in the main text in sufficient detail.

This is a good point, which we have addressed by moving the detailed explanation of the method to the main Materials and methods section. Additionally, we provide a summary explanation in the Results.

5) In the manuscript, there is a confusion between tools and methods. A software tool can be the concrete implementation of one method, but in most cases, a single tool (such as Ilastik) contains a range of methods. The authors should refer to both tool and method, in particular when they discuss benchmarking. In particular, they compare detection with segmentation methods, which is not rigorous.

This is a good point, which we have addressed by referring appropriately, in the manuscript text and figure legends, to methods (the Ilastik Pixel Classification method, and the CytoCensus method), and to software tools (Ilastik, CytoCensus).

[Editors' note: further revisions were suggested prior to acceptance, as described below.]Reviewer #2:[…]1) As mentioned in my previous review, I still do not agree with the way the results are presented in Figure 3: the authors compare their results to Ilastik. Concretely, they use Ilastik to segment cells (via pixel-classification). In a second step, they calculate the cell centers from this segmentation result as the centers of the connected components. An example of this is shown in Figure 3B' (2 quotes, upper right) and the corresponding statistics in Figure 3B' (4 quotes, bottom left). The problem with this benchmark is that this is not a reasonable approach for cell center detection. For this reason, I feel that the comparison is misleading and should not be published in this way. Normally, one would always try to apply at least some simple postprocessing for object splitting, prior to calculating the centers. The authors did this additional analysis (cited in the text) in the revised version of the manuscript, and achieved 0.88 ± 0.09 vs. 0.98 ± 0.05 with CytoCensus. This is the proper result to be reported. The difference is less striking, but it is also more realistic and will ultimately convince the readers more than the version that is currently in the manuscript. I therefore require that the Figure 3B' (4 quotes, bottom left) and Figure 3B' (2 quotes, upper right) are replaced by the corresponding figures from this more realistic scenario. The comparison to Ilastik pixel classification without post-processing should be removed.

We have made the changes that reviewer 2 has requested, but would like to note the following:

We agree with the reviewer that (from an image analysis perspective) our comparison to Ilastik with post-processing is fairer and more representative of typical analysis pipelines performed by those with image analysis experience, than a comparison of the direct outputs of Ilastik and CytoCensus by a more naïve user of the software. We have therefore made the changes the reviewer has requested. However, we would like to point out that our original comparison (to Ilastik without post-processing, F1-score: 0.21 ± 0.13) is more representative of how a naïve user without prior image analysis experience might approach the problem.

2) Regarding the results presented in Figure 3—figure supplement 2, the authors should also provide the rank of their method (ranked x out of N participants).

We have made the changes that reviewer 2 has suggested. Please refer to the revised Figure 3,Figure 3—figure supplement 2 and the manuscript text for the exact changes.